# DIFFUSION MODELS ARE REAL-TIME GAME ENGINES

**Dani Valevski**[*]
Google Research
daniv@google.com

**Yaniv Leviathan**[*]
Google Research
leviathan@google.com

**Moab Arar**[*]
Tel Aviv University[†]
moab.arar@gmail.com

**Shlomi Fruchter**[*]
Google DeepMind
shlomif@google.com

## ABSTRACT

We present *GameNGen*, the first game engine powered entirely by a neural model that also enables real-time interaction with a complex environment over long trajectories at high quality. When trained on the classic game DOOM, GameNGen extracts gameplay and uses it to generate a playable environment that can interactively simulate new trajectories. *GameNGen* runs at 20 frames per second on a single TPU and remains stable over extended multi-minute play sessions. Next frame prediction achieves a PSNR of 29.4, comparable to lossy JPEG compression. Human raters are only slightly better than random chance at distinguishing short clips of the game from clips of the simulation, even after 5 minutes of auto-regressive generation. *GameNGen* is trained in two phases: (1) an RL-agent learns to play the game and the training sessions are recorded, and (2) a diffusion model is trained to produce the next frame, conditioned on the sequence of past frames and actions. Conditioning augmentations help ensure stable auto-regressive generation over long trajectories, and decoder fine-tuning improves the fidelity of visual details and text.

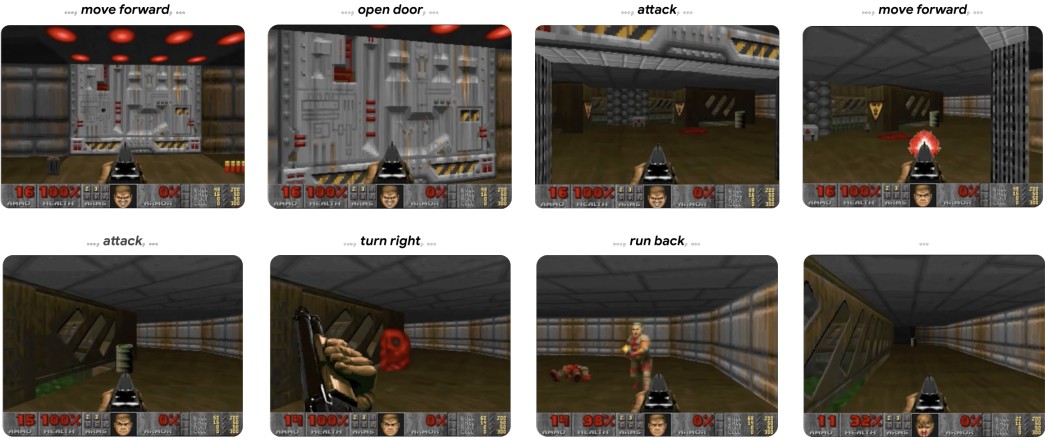

Figure 1: A human player is playing DOOM on **GameNGen** at 20 FPS. See supplementary material for multi-minute real-time video recordings of people interactively playing with GameNGen.

## 1 INTRODUCTION

Computer games are manually crafted software systems centered around the following *game loop*: (1) update the game state based on user input, and (2) render the game state to screen pixels. This game loop, running at high frame rates, creates the illusion of an interactive virtual world for the player. Such game loops are classically run on standard computers, and while there have been many impressive attempts at running games on bespoke hardware (e.g. the iconic game DOOM has been run on kitchen appliances, a treadmill, a camera, and within the game of Minecraft[1]), in all of these

---

[*]Equal contribution.

[†]Work done while at Google Research.

[1]See https://www.reddit.com/r/itrunsdoom/

cases the hardware is still emulating the manually written game software as-is. Furthermore, while vastly different game engines exist, the game state updates and rendering logic in all are composed of a set of manual rules, programmed or configured by hand.

In recent years, generative models made significant progress in producing images and videos conditioned on multi-modal inputs, such as text or images. At the forefront of this wave, diffusion models became the de-facto standard in media (i.e. non-language) generation, with works like Dall-E (Ramesh et al., 2022), Stable Diffusion (Rombach et al., 2022) and Sora (Brooks et al., 2024). At a glance, simulating the interactive worlds of video games may seem similar to video generation. However, *interactive* world simulation is more than just very fast video generation. The requirement to condition on a stream of input actions that is only available throughout the generation breaks some assumptions of existing diffusion model architectures. Notably, it requires generating frames autoregressively which tends to be unstable and leads to sampling divergence (see Section 3.2.1).

Several important works (Ha & Schmidhuber, 2018; Kim et al., 2020; Bruce et al., 2024) (see Section 6) simulate interactive video games with neural models. Nevertheless, most of these approaches are limited in respect to the complexity of the simulated games, simulation speed, stability over long time periods, or visual quality (see Figure 2). It is therefore natural to ask:

*Can a neural model running in real-time simulate a complex game at high quality?*

In this work we demonstrate that the answer is yes. Specifically, we show that a complex video game, the iconic game DOOM, can be run on a neural network, an augmented version of the open Stable Diffusion v1.4 (Rombach et al., 2022), in real-time, while achieving a visual quality comparable to that of the original game. While not an exact simulation (see limitations in Section 7), the neural model is able to perform complex game state updates, such as tallying health and ammo, attacking enemies, damaging objects, opening doors, and more generally persist the game state over long trajectories.

Our key contribution is a demonstration that a complex video game (DOOM) can be simulated by a neural network in real time with high quality on a single TPU. We provide concrete architecture and technical insights on how to (1) adapt a text-to-image diffusion model in a stable auto-regressive setup via noise augmentation, (2) achieve visual quality comparable to the original via fine-tuning the latent decoder, and (3) collect training data from an existing game at scale via an RL agent.

More broadly, demonstrating that real-time simulation of complex games on existing hardware is possible addresses one important question on the path towards a new paradigm for game engines – one where games are automatically generated, much like how images and videos have been generated by neural models in recent years. While bigger questions remain, such as how to use human input to create entirely new games instead of simulating existing ones, we are nevertheless excited for the possibilities of this new paradigm (see Section 7 for further discussion).

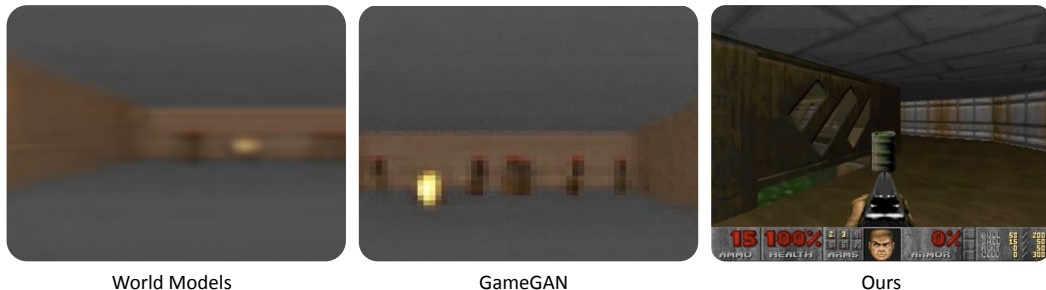

| World Models | GameGAN | Ours |

Figure 2: **GameNGen compared to prior simulations of DOOM**: World Models Ha & Schmidhuber (2018) and GameGAN Kim et al. (2020). Note that prior models are trained on different data.

## 2   INTERACTIVE WORLD SIMULATION

An *Interactive Environment* $\mathcal{E}$ consists of a space of latent states $\mathcal{S}$, a space of observations of the latent space $\mathcal{O}$, a partial projection function $V : \mathcal{S} \to \mathcal{O}$, a set of actions $\mathcal{A}$, and a transition probability function $p(s|a, s')$ such that $s, s' \in \mathcal{S}, a \in \mathcal{A}$.

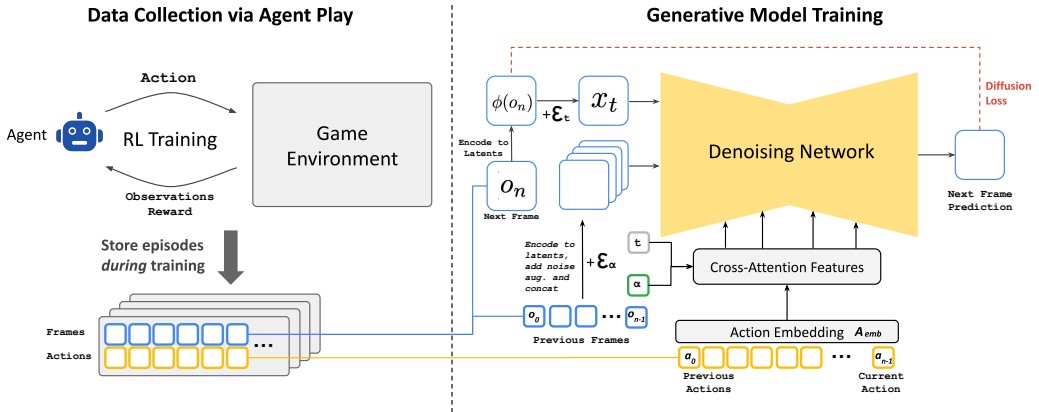

Figure 3: Method overview (see Section 3).

For example, in the case of the game DOOM, $\mathcal{S}$ is the program's dynamic memory contents, $\mathcal{O}$ is the rendered screen pixels, $V$ is the game's rendering logic, $\mathcal{A}$ is the set of key presses, and $p$ is the program's logic given the player's input (including any potential non-determinism).

Given an input interactive environment $\mathcal{E}$, and an initial state $s_0 \in \mathcal{S}$, an *Interactive World Simulation* is a *simulation distribution function* $q(o_n|o_{<n}, a_{\leq n}), o_i \in \mathcal{O}, a_i \in \mathcal{A}$. Given a distance metric between observations $D : \mathcal{O} \times \mathcal{O} \to \mathbb{R}$, a *policy*, i.e. a distribution on agent actions given past actions and observations $\pi(a_n|o_{<n}, a_{<n})$, a distribution $S_0$ on initial states, and a distribution $N_0$ on episode lengths, the *Interactive World Simulation* objective consists of minimizing $E(D(o_q^i, o_p^i))$ where $n \sim N_0$, $0 \leq i \leq n$, and $o_q^i \sim q, o_p^i \sim V(p)$ are sampled observations from the environment and the simulation when enacting the agent's policy $\pi$. Importantly, the conditioning actions for these samples are always obtained by the agent interacting with the environment $\mathcal{E}$, while the conditioning observations can either be obtained from $\mathcal{E}$ (the *teacher forcing objective*) or from the simulation (the *auto-regressive objective*).

We always train our generative model with the teacher forcing objective. Given a simulation distribution function $q$, the environment $\mathcal{E}$ can be simulated by auto-regressively sampling observations.

## 3    GAMENGEN

GameNGen (pronounced "game engine") is a generative diffusion model that learns to simulate the game under the settings of Section 2. In order to collect training data for this model at scale, with the teacher forcing objective, we first train a separate model to interact with the environment. The two models (agent and generative) are trained in sequence. The entirety of the agent's actions and observations corpus $\mathcal{T}_{agent}$ during training is maintained and becomes the training dataset for the generative model in a second stage. See Figure 3.

### 3.1    DATA COLLECTION VIA AGENT PLAY

Our end goal is to have human players interact with our simulation. To that end, the policy $\pi$ as in Section 2 is that of *human gameplay*. Since we cannot sample from that directly at scale, we start by approximating it via teaching an automatic agent to play. Unlike a typical RL setup which attempts to maximize game score, our goal is to generate training data which resembles human play, or at least contains enough diverse examples, in a variety of scenarios, to maximize training data efficiency. To that end, we design a simple reward function, which is the only part of our method that is environment-specific (see Appendix A.5).

We record the agent's training trajectories throughout the entire training process, which includes different skill levels of play, starting with a random policy when the agent is untrained. This set of recorded trajectories is our $\mathcal{T}_{agent}$ dataset, used for training the generative model (see Section 3.2).

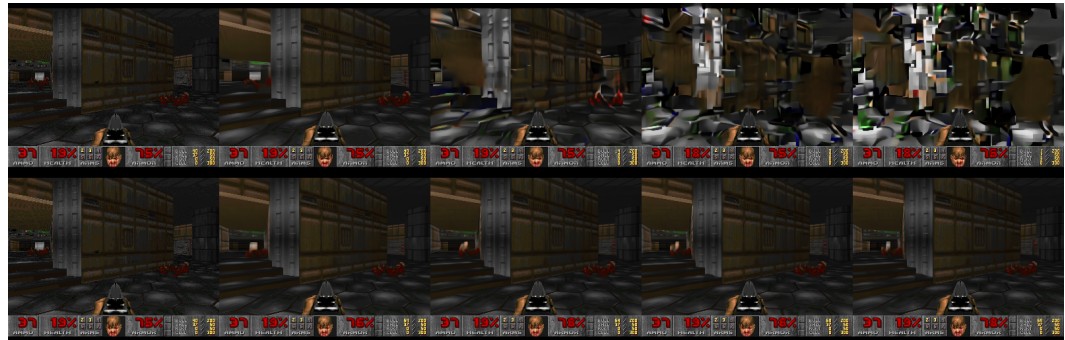

Figure 4: **Auto-regressive drift.** Top: we present every 10th frame of a simple trajectory with 50 frames in which the player is not moving. Quality degrades fast after 20-30 steps. Bottom: the same trajectory with noise augmentation does not suffer from quality degradation.

## 3.2 TRAINING THE GENERATIVE DIFFUSION MODEL

We now train a generative diffusion model conditioned on the agent's trajectories $\mathcal{T}_{agent}$ (actions and observations) collected during the previous stage.

We re-purpose a pre-trained text-to-image diffusion model, Stable Diffusion v1.4 (Rombach et al., 2022) to predict the next frame in the game. We condition the model $f_\theta$ on trajectories $T \sim \mathcal{T}_{agent}$, i.e. on a sequence of previous actions $a_{<n}$ and observations (frames) $o_{<n}$ and remove all text conditioning. Specifically, to condition on actions (i.e. key presses), we simply learn an embedding $A_{emb}$ from each action into a single token and replace the cross attention from the text into this encoded actions sequence. In order to condition on observations (i.e. previous frames) we encode them into latent space using the auto-encoder $\phi$ and concatenate them in the latent channels dimension to the noised latents (see Figure 3). We also experimented with conditioning on these past observations via cross-attention but observed no meaningful improvements.

We train the model to minimize the diffusion loss with velocity parameterization (Salimans & Ho, 2022):

$$\mathcal{L} = \mathbb{E}_{t,\epsilon,T} \left[ \|v(\epsilon, x_0, t) - v_{\theta'}(x_t, t, \{\phi(o_{i<n})\}, \{A_{emb}(a_{i<n})\})\|_2^2 \right] \quad (1)$$

where $T = \{o_{i\leq n}, a_{i\leq n}\} \sim \mathcal{T}_{agent}$, $t \sim \mathcal{U}(0,1)$, $\epsilon \sim \mathcal{N}(0, \mathbf{I})$, $x_t = \sqrt{\bar{\alpha}_t}x_0 + \sqrt{1 - \bar{\alpha}_t}\epsilon$, $x_0 = \phi(o_n)$, $v(\epsilon, x_0, t) = \sqrt{\bar{\alpha}_t}\epsilon - \sqrt{1 - \bar{\alpha}_t}x_0$, and $v_{\theta'}$ is the v-prediction output of the model $f_\theta$. The noise schedule $\bar{\alpha}_t$ is linear, similarly to Rombach et al. (2022).

### 3.2.1 MITIGATING AUTO-REGRESSIVE DRIFT USING NOISE AUGMENTATION

The domain shift between training with teacher-forcing and auto-regressive sampling leads to error accumulation and fast degradation in sample quality, as demonstrated in Figure 4 (top). To avoid this divergence due to auto-regressive application of the model, we corrupt context frames by adding a varying amount of Gaussian noise to encoded frames in training time, while providing the noise level as input to the model, following Ho et al. (2021). To that effect, we sample a noise level $\alpha$ uniformly up to a maximal value, discretize it and learn an embedding for each bucket (see Figure 3). This allows the network to correct information sampled in previous frames, and is critical for preserving frame quality over time. During inference, the added noise level can be controlled to maximize quality, although we find that even with no added noise the results are significantly improved. We ablate the impact of this method in Section 5.2.2.

### 3.2.2 LATENT DECODER FINE-TUNING

The pre-trained auto-encoder of Stable Diffusion v1.4, which compresses 8x8 pixel patches into 4 latent channels, results in meaningful artifacts when predicting game frames, which affect small details and particularly the bottom bar HUD ("heads up display"). To hopefully leverage some of

the pre-trained knowledge while improving image quality, we train just the decoder of the latent auto-encoder using an MSE loss computed against the target frame pixels. Importantly, note that this fine tuning process happens completely separately from the U-Net fine-tuning, and that notably the auto-regressive generation isn't affected by it (we only condition auto-regressively on the latents, not the pixels). Appendix A.2 shows examples of generations with and without fine-tuning the auto-encoder.

## 3.3 INFERENCE

### 3.3.1 SETUP

We use DDIM sampling (Song et al., 2022). We employ Classifier-Free Guidance (Ho & Salimans, 2022) only for the past observations condition $o_{<n}$. We didn't find guidance for the past actions condition $a_{<n}$ to improve quality. The weight we use is relatively small (1.5) as larger weights create artifacts which increase due to our auto-regressive sampling.

### 3.3.2 DENOISER SAMPLING STEPS

During inference, we need to run both the U-Net denoiser (for a number of steps) and the auto-encoder. On our hardware configuration (a single TPU-v5), a single denoiser step and an evaluation of the auto-encoder both takes 10ms. If we ran our model with a single denoiser step, the minimum total latency possible in our setup would therefore be 20ms per frame, or 50 frames per second. Usually, generative diffusion models, such as Stable Diffusion, don't produce high quality results with a single denoising step, and instead require dozens of sampling steps to generate a high quality image. Surprisingly, we found that we can robustly simulate DOOM, with only 4 DDIM sampling steps (Song et al., 2020). In fact, we observe no degradation in simulation quality when using 4 sampling steps vs 20 steps or more (see Table 1 and Appendix A.6). Using just 4 denoising steps

Table 1: **Generation with Varying Sampling Steps.** We evaluate the generation quality of a GameNGen model with an increasing number of steps using PSNR and LPIPS metrics. "D" marks a 1-step distilled model. See Appendix A.6 for more details.

| Steps | PSNR ↑ | LPIPS ↓ |
|---|---|---|
| D | $31.10 \pm 0.098$ | $0.208 \pm 0.002$ |
| 1 | $25.47 \pm 0.098$ | $0.255 \pm 0.002$ |
| 2 | $31.91 \pm 0.104$ | $0.205 \pm 0.002$ |
| 4 | $32.58 \pm 0.108$ | $0.198 \pm 0.002$ |
| 8 | $32.55 \pm 0.110$ | $0.196 \pm 0.002$ |
| 16 | $32.44 \pm 0.110$ | $0.196 \pm 0.002$ |
| 32 | $32.32 \pm 0.110$ | $0.196 \pm 0.002$ |
| 64 | $32.19 \pm 0.110$ | $0.197 \pm 0.002$ |

leads to a total U-Net cost of 40ms (and total inference cost of 50ms, including the auto encoder) or 20 frames per second. We hypothesize that the negligible impact to quality with few steps in our case stems from a combination of: (1) a constrained images space, and (2) strong conditioning by the previous frames.

Since we do observe degradation when using just a single sampling step, we also experimented with model distillation similarly to (Yin et al., 2024; Wang et al., 2023) in the single-step setting. Distillation does help substantially there (allowing us to reach 50 FPS as above), but still comes at a some cost to simulation quality, so we opt to use the 4-step version without distillation for our method (see Appendix A.6).

## 4 EXPERIMENTAL SETUP

### 4.1 AGENT TRAINING

The agent model is trained using PPO (Schulman et al., 2017), with a simple CNN as the feature network, following Mnih et al. (2015b). It is trained on CPU using the Stable Baselines 3 infras-

tructure (Raffin et al., 2021). The agent is provided with downscaled versions of the frame images and in-game map, each at resolution 160x120. The agent also has access to the last 32 actions it performed. The feature network computes a representation of size 512 for each image. PPO's actor and critic are 2-layer MLP heads on top of a concatenation of the outputs of the image feature network and the sequence of past actions. We train the agent to play the game using the ViZDoom environment (Wydmuch et al., 2019). We run 8 games in parallel, each with a replay buffer size of 512, a discount factor $\gamma = 0.99$, and an entropy coefficient of $0.1$. In each iteration, the network is trained using a batch size of 64 for 10 epochs, with a learning rate of 1e-4. We perform a total of 50M environment steps.

## 4.2 GENERATIVE MODEL TRAINING

We train all simulation models from a pretrained checkpoint of Stable Diffusion 1.4, unfreezing all U-Net parameters. We use a batch size of 128 and a constant learning rate of 2e-5, with the Adafactor optimizer without weight decay (Shazeer & Stern, 2018) and gradient clipping of 1.0. The context frames condition is dropped with probability 0.1 to allow CFG during inference. We train using 128 TPU-v5e devices with data parallelization. Unless noted otherwise, all results in the paper are after 700,000 training steps. For noise augmentation (Section 3.2.1), we use a maximal noise level of 0.7, with 10 embedding buckets. We use a batch size of 2,048 for optimizing the latent decoder, other training parameters are identical to those of the denoiser. For training data, we use a random subset of 70M examples from the recorded trajectories played by the agent during RL training and evaluation (see Appendix A.3 for results with smaller datasets). All image frames (during training, inference, and conditioning) are at a resolution of 320x240 padded to 320x256. We use a context length of 64 (i.e. the model is provided its own last 64 predictions as well as the last 64 actions).

## 5 RESULTS

### 5.1 SIMULATION QUALITY

Overall, our method achieves a simulation quality comparable to the original game over long trajectories in terms of image quality. Human raters are only slightly better than random chance at distinguishing between short clips of the simulation and the actual game.

**Image Quality.** We measure LPIPS (Zhang et al., 2018) and PSNR using the teacher-forcing setup described in Section 2, where we sample an initial state and predict a single frame based on a trajectory of ground-truth past observations. When evaluated over a random holdout of 2048 trajectories taken in 5 different levels, our model achieves a PSNR of $29.43$ and an LPIPS of $0.249$. The PSNR value is similar to lossy JPEG compression with quality settings of 20-30 (Petric & Milinkovic, 2018). Figure 5 shows examples of model predictions and the corresponding ground truth samples.

**Video Quality.** We use the auto-regressive setup described in Section 2, where we iteratively sample frames following the sequences of actions defined by the ground-truth trajectory, while conditioning the model on its own past predictions. When sampled auto-regressively, the predicted and ground-truth trajectories often diverge after a few steps, mostly due to the accumulation of small amounts of different movement velocities between frames in each trajectory. For that reason, per-frame PSNR and LPIPS values gradually decrease and increase respectively, as can be seen in Figure 6. The predicted trajectory is still similar to the actual game in terms of content and image quality, but per-frame metrics are limited in their ability to capture this (see Appendix A.1 for samples of auto-regressively generated trajectories).

We therefore measure the FVD (Unterthiner et al., 2019) computed over a random holdout of 512 trajectories, measuring the distance between the predicted and ground truth trajectory distributions, for simulations of length 16 frames (0.8 seconds) and 32 frames (1.6 seconds). For 16 frames our model obtains an FVD of $114.02$. For 32 frames our model obtains an FVD of $186.23$.

**Human Evaluation.** As another measurement of simulation quality, we provided 10 human raters with 130 random short clips (of lengths 1.6 seconds and 3.2 seconds) of our simulation side by side with the real game. The raters were tasked with recognizing the real game (see Figure 17 in Appendix A.8). The raters only choose the actual game over the simulation in 58% or 60% of the time (for the 1.6 seconds and 3.2 seconds clips, respectively). To evaluate the impact of accumulated

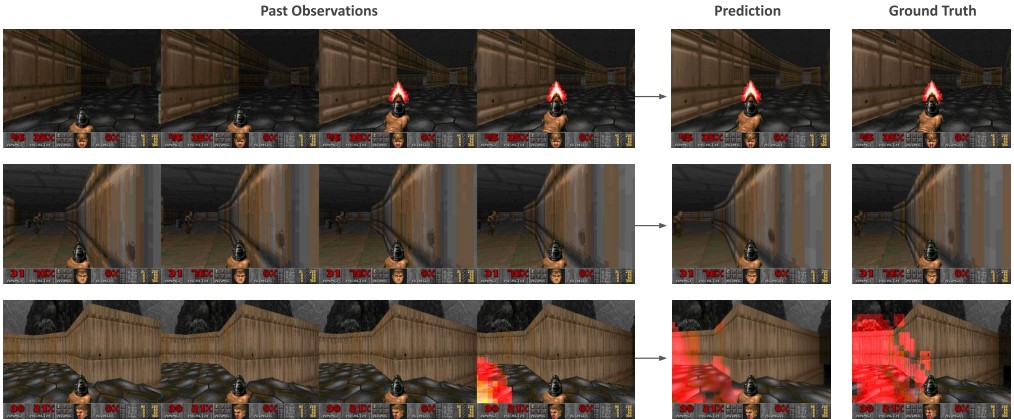

Figure 5: **Model predictions vs. ground truth.** Only the last 4 frames of the past observations context are shown.

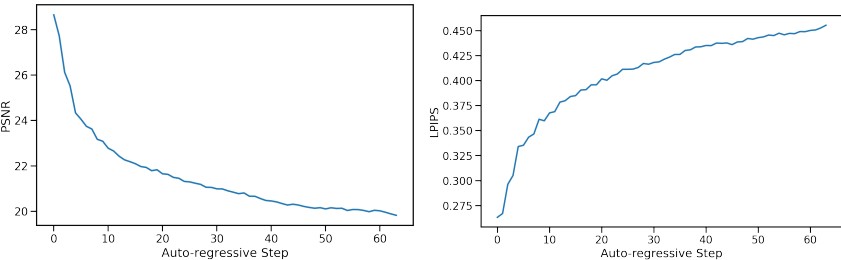

Figure 6: **Auto-regressive evaluation.** PSNR and LPIPS metrics over 64 auto-regressive steps.

errors from auto-regressive generation, we generated 150 additional side-by-side comparisons using clips of 3 seconds in length after 5 to 10 minutes of gameplay. Despite the extended period of auto-regressive generation, raters still performed at chance level, identifying the real game only 50% of the time. While human raters struggled to distinguish between the simulation and real gameplay in short clips, it is important to note that the authors, who are familiar with the specific limitations of the simulation, could often identify the real game after just a few seconds of play. For qualitative assessment of longer multi-minute clips, refer to the videos in the supplementary material.

## 5.2 ABLATIONS

To evaluate the importance of the different components of our methods, we sample trajectories from the evaluation dataset and compute LPIPS and PSNR metrics between the ground truth and the predicted frames.

### 5.2.1 CONTEXT LENGTH

We evaluate the impact of changing the number $N$ of past observations in the conditioning context by training models with $N \in \{1, 2, 4, 8, 16, 32, 64\}$ (recall that our method uses $N = 64$). This affects both the number of historical frames and actions. We train the models for 200,000 steps keeping the decoder frozen and evaluate on test-set trajectories from 5 levels. See the results in Table 2. As expected, we observe that generation quality improves with the length of the context. Interestingly, we observe that while the improvement is large at first (e.g. between 1 and 2 frames), we quickly approach an asymptote and further increasing the context size provides only small improvements in quality. This is somewhat surprising as even with our maximal context length, the model only has access to a little over 3 seconds of history. Notably, we observe that much of the game state is persisted for much longer periods (see Section 7). While the length of the conditioning context

is an important limitation, Table 2 hints that we'd likely need to change the architecture or training scheme of our model to efficiently support longer contexts, and employ better selection of the past frames to condition on, which we leave for future work.

Table 2: **Number of history frames.** We ablate the number of history frames used as context using 8912 test-set examples from 5 levels. More frames generally improve both PSNR and LPIPS metrics.

| History Context Length | PSNR ↑ | LPIPS ↓ |
|---|---|---|
| 64 | $22.36 \pm 0.033$ | $0.295 \pm 0.001$ |
| 32 | $22.31 \pm 0.033$ | $0.296 \pm 0.001$ |
| 16 | $22.28 \pm 0.033$ | $0.296 \pm 0.001$ |
| 8 | $22.26 \pm 0.033$ | $0.296 \pm 0.001$ |
| 4 | $22.26 \pm 0.034$ | $0.298 \pm 0.001$ |
| 2 | $22.03 \pm 0.037$ | $0.304 \pm 0.001$ |
| 1 | $20.94 \pm 0.044$ | $0.358 \pm 0.001$ |

### 5.2.2 NOISE AUGMENTATION

To ablate the impact of noise augmentation we train a model without added noise. We evaluate both our standard model with noise augmentation and the model without added noise (after 200k training steps) auto-regressively and compute PSNR and LPIPS metrics between the predicted frames and the ground-truth over a random holdout of 512 trajectories. We report average metric values for each auto-regressive step up to a total of 64 frames in Figure 7.

Without noise augmentation, LPIPS distance from the ground truth increases rapidly compared to our standard noise-augmented model, while PSNR drops, indicating a divergence of the simulation from ground truth.

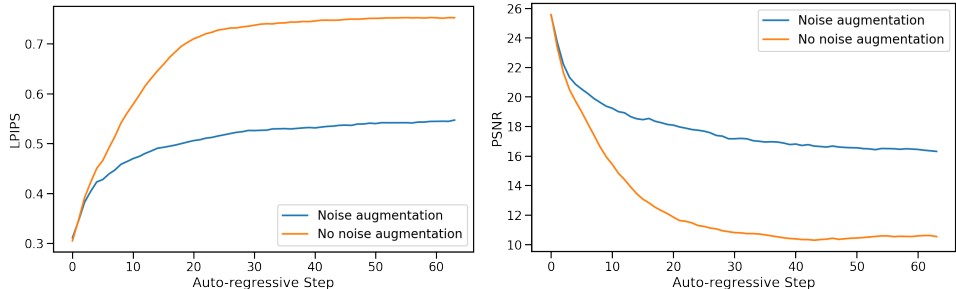

Figure 7: **Impact of Noise Augmentation.** The plots show average LPIPS (lower is better) and PSNR (higher is better) values for each auto-regressive step. When noise augmentation is not used quality degrades quickly after 10-20 frames. This is prevented by noise augmentation.

### 5.2.3 AGENT PLAY

We compare training on agent-generated data to training on data generated using a random policy. For the random policy, we sample actions following a uniform categorical distribution that doesn't depend on the observations. We compare the random and agent datasets by training 2 models for 700k steps along with their decoder. The models are evaluated on a dataset of 2048 human-play trajectories from 5 levels. We compare the first frame of generation, conditioned on a history context of 64 ground-truth frames, as well as a frame after 3 seconds of auto-regressive generation.

Overall, we observe that training the model on random trajectories works surprisingly well, but is limited by the exploration ability of the random policy. When comparing the single frame generation the agent works only slightly better, achieving a PNSR of 25.06 vs 24.42 for the random policy. When comparing a frame after 3 seconds of auto-regressive generation, the difference increases to 19.02 vs 16.84. When playing with the model manually, we observe that some areas are very easy for both, some areas are very hard for both, and in some the agent performs much better. With that, we manually split 456 examples into 3 buckets: easy, medium, and hard, manually, based on their

Table 3: **Performance on Different Difficulty Levels.** We compare the performance of models trained using Agent-generated and Random-generated data across easy, medium, and hard splits of the dataset. Easy and medium have 112 items, hard has 232 items. Metrics are computed for each trajectory on a single frame after 3 seconds.

| Difficulty Level | Data Generation Policy | PSNR ↑ | LPIPS ↓ |
|---|---|---|---|
| Easy | Agent | $20.94 \pm 0.76$ | $0.48 \pm 0.01$ |
| | Random | $20.20 \pm 0.83$ | $0.48 \pm 0.01$ |
| Medium | Agent | $20.21 \pm 0.36$ | $0.50 \pm 0.01$ |
| | Random | $16.50 \pm 0.41$ | $0.59 \pm 0.01$ |
| Hard | Agent | $17.51 \pm 0.35$ | $0.60 \pm 0.01$ |
| | Random | $15.39 \pm 0.43$ | $0.61 \pm 0.00$ |

distance from the starting position in the game. We observe that on the easy and hard sets, the agent performs only slightly better than random, while on the medium set the difference is much larger in favor of the agent as expected (see Table 3). See Figure 16 in Appendix A.7 for an example of the scores during a single session of human play.

## 6 RELATED WORK

**Game Simulation and World Models.** Several works attempted to train models for game simulation with actions inputs. Yang et al. (2023) build a diverse dataset of real-world and simulated videos and train a diffusion model to predict a continuation video given a previous video segment and a textual description of an action. Menapace et al. (2021) and Bruce et al. (2024) focus on unsupervised learning of actions from videos. Menapace et al. (2024) converts textual prompts to game states, which are later converted to a 3D representation using NeRF. Another line of work explored learning a predictive model of the environment and using it for training an RL agent. Ha & Schmidhuber (2018) train a Variational Auto-Encoder (Kingma & Welling, 2014) to encode game frames into a latent vector, and then use an RNN to mimic the ViZDoom game environment, training on random rollouts from a random policy (i.e. selecting an action at random). Then controller policy is learned by playing within the "hallucinated" environment. Hafner et al. (2020) demonstrate that an RL agent can be trained entirely on episodes generated by a learned world model in latent space.

Also close to our work is Kim et al. (2020), that use an LSTM architecture for modeling the world state, coupled with a convolutional decoder for producing output frames and jointly trained under an adversarial objective. Yan et al. (2021) apply a decoder-only transformer architecture to video generation using a discrete latent space learned by a VQ-VAE, and show that it can be used for action-conditioned video generation on the VizDoom simulator. Hu et al. (2023) focuses on the problem of action-conditioned world modeling in the domain of driving. A multi-modal autoregressive transformer acts as the world model and predicts the latent-space image token based on past image, text and action tokens. Then a diffusion video decoder translates the image tokens into a pixel-space video. The models are trained on large corpus of real-work driving data (420M unique images). Finally, concurrently with our work, Alonso et al. (2024) train a diffusion world model to predict the next observation given observation history, and iteratively train the world model and an RL model on Atari games. More recent version of their work also included a high-res simulation of Counter-Strike, trained on 95 hours of human game play recording.

**Auto-regressive Diffusion Models**. Some recent work explored auto-regressive architectures of diffusion models. Chen et al. (2024) diverge from traditional diffusion architectures by allowing each token to have its own level of noise in each time step, reporting this approach to improve stability of video generation beyond the training horizon, when using a convolutional RNN backbone. Ruhe et al. (2024) also allow variable levels of noise for different tokens, using a sliding window denoising process. Such approaches are interesting to explore for real-time game simulation, and we leave this to future work.

**DOOM.** When DOOM was released in 1993 it revolutionized the gaming industry. Introducing groundbreaking 3D graphics technology, it became a cornerstone of the first-person shooter genre,

influencing countless other games. DOOM was studied by numerous research works. It provides an open-source implementation and a native resolution that is low enough for small sized models to simulate, while being complex enough to be a challenging test case. Finally, the authors have spent countless youth hours with the game. It was a trivial choice to use it in this work.

## 7 DISCUSSION

**Summary.** We introduced *GameNGen*, and demonstrated that high-quality real-time gameplay at 20 frames per second is possible on a neural model.

**Limitations.** (1) GameNGen suffers from a limited amount of memory. The model only has access to a little over 3 seconds of history, so it's remarkable that much of the game logic is persisted for drastically longer time horizons (see the supplementary for multi-minute gameplay). While some of the game state is persisted through screen pixels (e.g. ammo and health tallies, available weapons, etc.), the model likely learns strong heuristics that allow meaningful generalizations. For example, the model infers the current location from the rendered view, and can guess if enemies in an area are defeated from the ammo and health tallies. That said, it's easy to create situations where this context length is not enough (see video in the supplementary). Also, these heuristics might be wrong, for example, if the player repeatedly shoots, GameNGen might decide to spawn an enemy, as the agent would usually shoot when enemies are present. Continuing to increase the context size with our existing architecture yields only marginal benefits (Section 5.2.1), and the model's short context length remains an important limitation. (2) The second important limitation are the remaining differences between the agent's behavior and those of human players. For example, our agent, even at the end of training, still does not explore all of the game's locations and interactions, leading to erroneous behavior in those cases (see video in the supplementary). (3) Another important limitation, is that we are not able to easily produce new games with GameNGen. Like traditional game engines, GameNGen interactively runs the game-loop (update state based on input and render it to the screen). However, most game engines offer another important feature which is the ability to *easily* create new games, which GameNGen currently lacks.

**Future Work.** We demonstrate GameNGen on the classic game DOOM. It would be interesting to test it on other games or more generally on other interactive software systems; We note that nothing in our technique is DOOM specific except for the reward function for the RL-agent. We plan on addressing that in a future work; While GameNGen manages to maintain game state accurately, it isn't perfect, as per the discussion above. A more sophisticated architecture and training scheme might be needed to mitigate these; GameNGen currently has a limited capability to leverage more than a minimal amount of memory. Experimenting with further expanding the memory effectively could be critical for more complex games/software; GameNGen runs at 20 or 50 FPS[2] on a TPUv5. It would be interesting to experiment with further optimization techniques to get it to run at higher frame rates and on consumer hardware.

**Towards a New Paradigm for Interactive Video Games.** Today, video games are *programmed* by humans. GameNGen is a proof-of-concept for one part of a new paradigm where games are weights of a neural model, not lines of code. GameNGen shows that an architecture and model weights exist such that a neural model can effectively run a complex game (DOOM) interactively on existing hardware. While many important questions remain, we are hopeful that this paradigm could have important benefits. For example, the development process for video games under this new paradigm might be less costly and more accessible, whereby games could be developed and edited via textual descriptions or examples images. A small part of this vision, namely creating modifications or novel behaviors for existing games, might be achievable in the shorter term. For example, we might be able to convert a set of frames into a new playable level or create a new character just based on example images, without having to author code (see Appendix A.4). Other advantages of this new paradigm include strong guarantees on frame rates and memory footprints. We have not experimented with these directions yet and much more work is required here, but we are excited to try! Hopefully this small step will someday contribute to a meaningful improvement in people's experience with video games, or maybe even more generally, in day-to-day interactions with interactive software systems.

---

[2]Faster than the original game DOOM ran on the some of the authors' 80386 machines at the time!

## BROADER IMPACT

**Societal impact.** GameNGen demonstrates that it is possible to simulate interactive games in real-time using neural networks, opening up new possibilities for game development. Similarly to other generative technologies like LLMs, text-to-image and text-to-video models, it will be important to explore how to empower users and game developers to build new experiences responsibly.

**Reproducibility.** We prioritized reproducibility in our implementation choices. We opt to use a relatively small open-source and open-weights foundation model (Stable Diffusion 1.4) which is widely accessible for fine-tuning. The game environment we use, VizDoom, is well-documented and open-source. Finally, we include detailed descriptions of training parameters and data generation configurations, and share performance metrics as a baseline for future work.

## ACKNOWLEDGEMENTS

We'd like to extend a huge thank you to Eyal Segalis, Eyal Molad, Matan Kalman, Nataniel Ruiz, Amir Hertz, Matan Cohen, Yossi Matias, Yael Pritch, Danny Lumen, Valerie Nygaard, Michelle Tadmor Ramanovich, the Theta Labs and Google Research teams, and our families for insightful feedback, ideas, suggestions, and support.

## CONTRIBUTION

Shlomi and Yaniv proposed the project, method, and architecture, developed the initial implementation, and made key contributions to implementation and writing. Dani developed much of the final codebase, tuned parameters and details across the system, added autoencoder fine-tuning, agent training, and distillation. Moab led auto-regressive stabilization with noise-augmentation, many of the ablations, and created the dataset of human-play data.

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

# A APPENDIX

## A.1 SAMPLES

Figs. 8,9,10,11 provide selected samples from GameNGen.

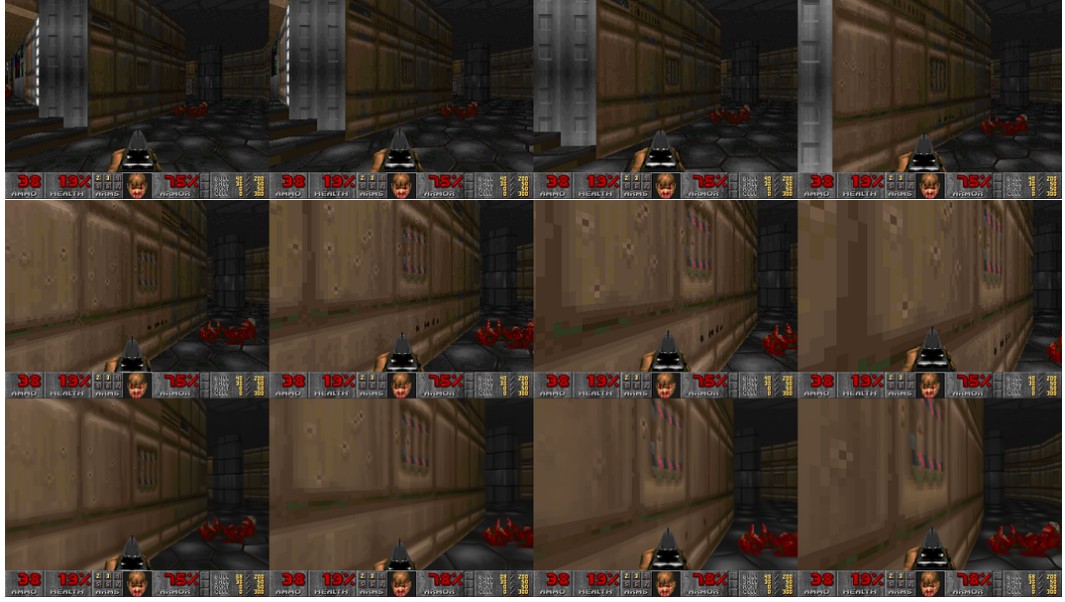

Figure 8: **Auto-regressive evaluation of the simulation model: Sample #1.** Top row: Context frames. Middle row: Ground truth frames. Bottom row: Model predictions.

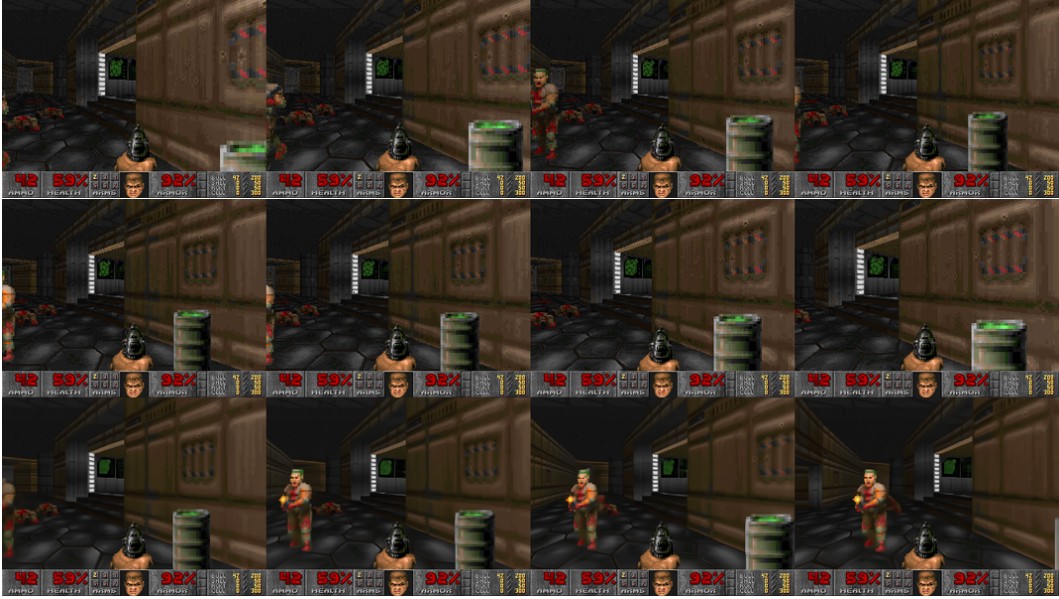

Figure 9: **Auto-regressive evaluation of the simulation model: Sample #2.** Top row: Context frames. Middle row: Ground truth frames. Bottom row: Model predictions.

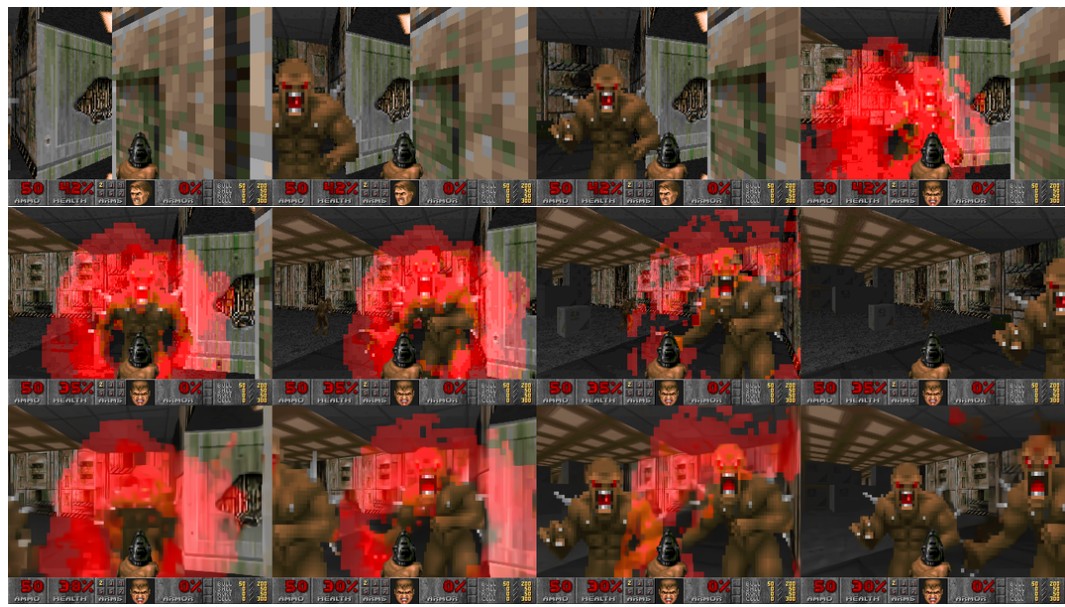

Figure 10: **Auto-regressive evaluation of the simulation model: Sample #3.** Top row: Context frames. Middle row: Ground truth frames. Bottom row: Model predictions.

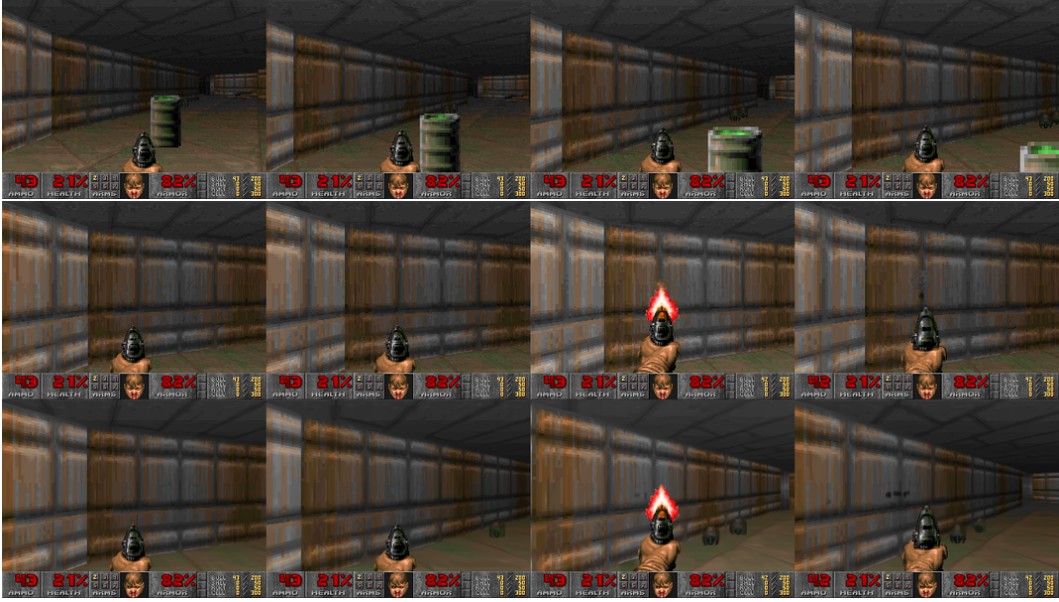

Figure 11: **Auto-regressive evaluation of the simulation model: Sample #4.** Top row: Context frames. Middle row: Ground truth frames. Bottom row: Model predictions.

## A.2 FINE-TUNING LATENT DECODER EXAMPLES

Fig. 12 demonstrates the effect of fine-tuning the vae decoder.

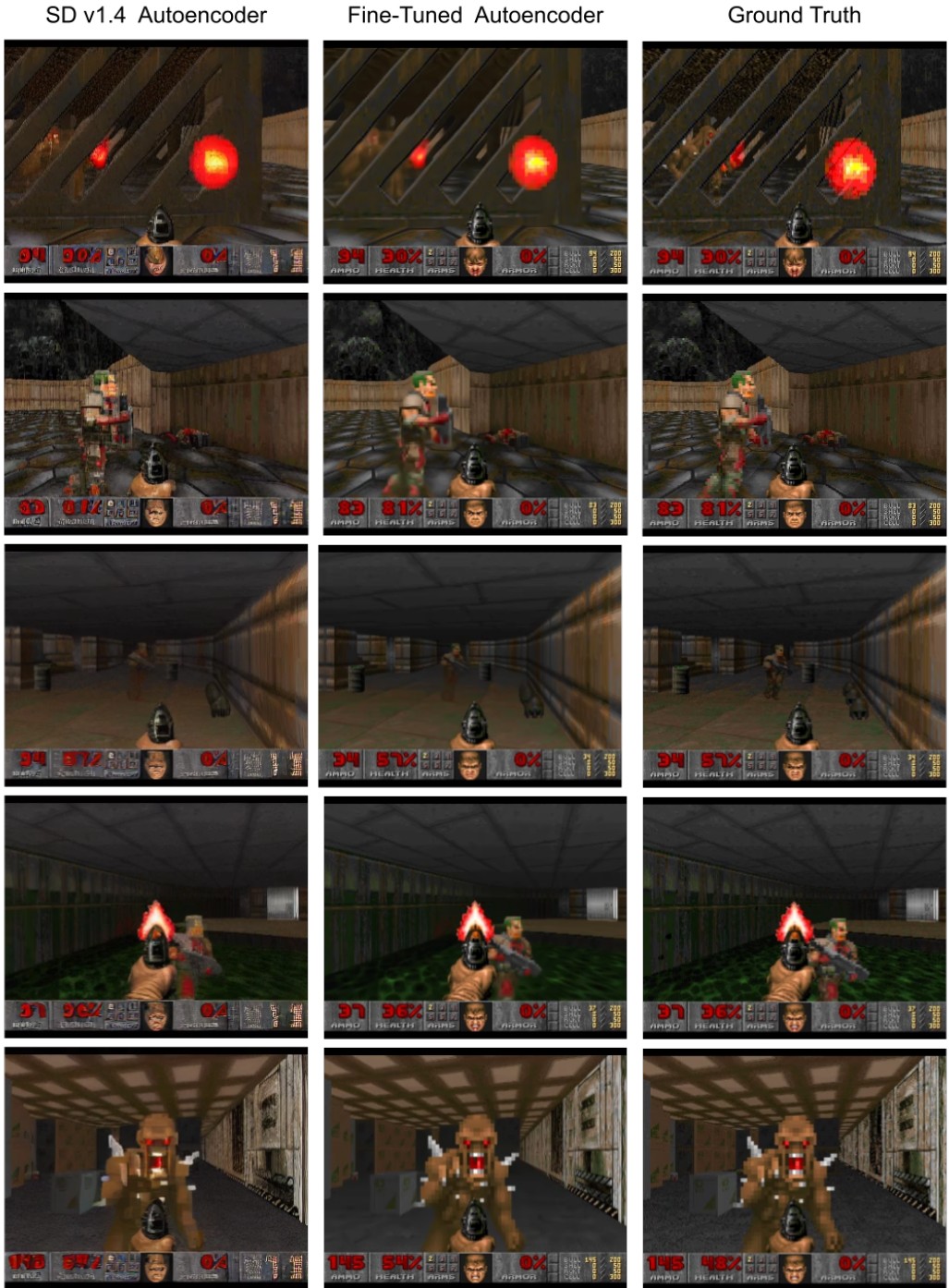

Figure 12: A comparison of generations with the standard latent decoder from Stable Diffusion v1.4 (Left), our fine-tuned decoder (Middle), and ground truth (Right). Artifacts in the frozen decoder are noticeable (e.g. in the numbers in the bottom HUD).

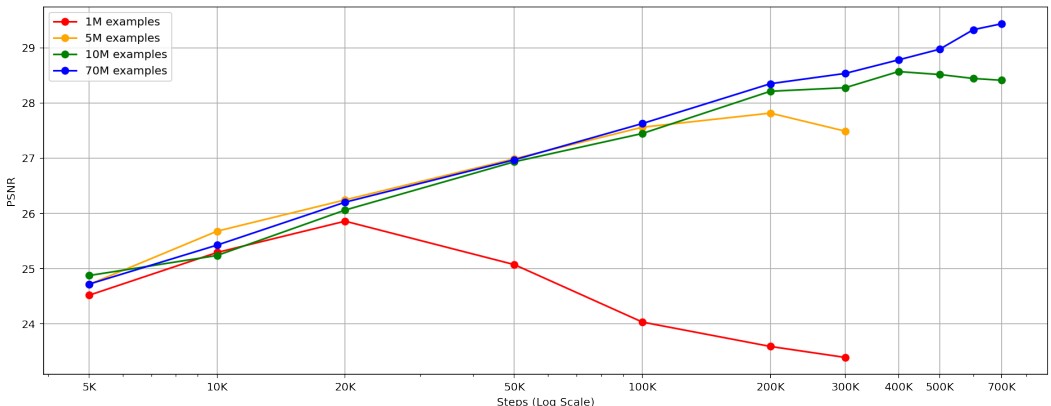

Figure 13: **Impact of dataset size.** PSNR vs. training step curves across different dataset sizes (1M, 5M, 10M, 70M examples), evaluated on 2048 unseen test trajectories. Note that given our batch size of 128, 10K steps correspond to a little over 1M examples.

### A.3 DATASET SIZE ANALYSIS

To evaluate the performance of GameNGen on datasets of different sizes, we trained three additional models with only 1M, 5M, and 10M examples. Fig. 13 shows the PSNR curves of models trained on these different datasets. PSNR for next frame prediction is evaluated on 2048 unseen trajectories from the test set. We observe that, as expected, for smaller datasets the test-set performance peaks earlier. For the 70M dataset, the performance continues to improve beyond 700k steps. As a qualitative measure, we also recorded human gameplay in models with 1M and 10M examples. For each model, we used the checkpoint with the lowest test-set loss (see files `condition{1,2}_{1M,10M}_examples.mp4` in the supplementary material). The model trained on 1M examples can render novel viewpoints well but struggles with consistency and game logic (it cannot kill monsters). With 10M examples, we observe improvements in detail and consistency.

### A.4 OUT-OF-DISTRIBUTION SAMPLING

To further explore GameNGen's ability to generate behaviors not present in the training data, we performed initial experiments by taking frames from the game, editing them with a graphical picture editor, and starting the generation from the edited frames. Specifically, we replicate the same frame for the entirety of the history buffer, with the "no key pressed" action. To encourage further generalization, in this setup we train a new GameNGen model where we randomize the agent's starting location. Due to the noise augmentation (Section 3.2.1), small local changes get ignored. When performing more substantial changes, the model usually generates unseen levels and situations. For example, Fig. 14 shows examples of pasting a game character into areas where they do not appear in the training data (e.g., inserting a monster from an advanced level into an early one). We observe that the model often consistently integrates the added characters into the new location, and they move, shoot at the player, and cause damage. Figure 15 demonstrates modifying a level's layout by inserting features such as walls, doors, or pools from other areas. The model successfully integrates these into the environment, rendering new viewpoints as the player navigates. We hope to further explore these preliminary results in future work.

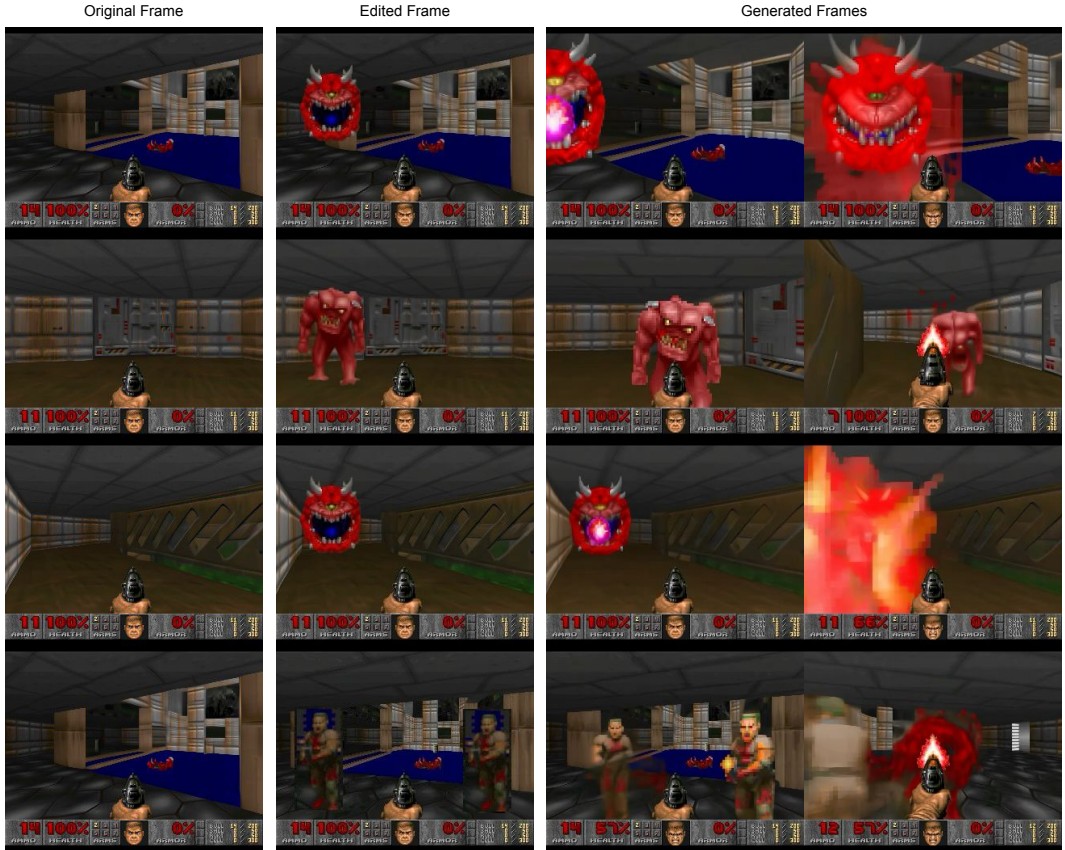

Figure 14: **Adding Characters.** Frames generated by GameNGen when starting generation from a manually edited state which includes characters from a different area.

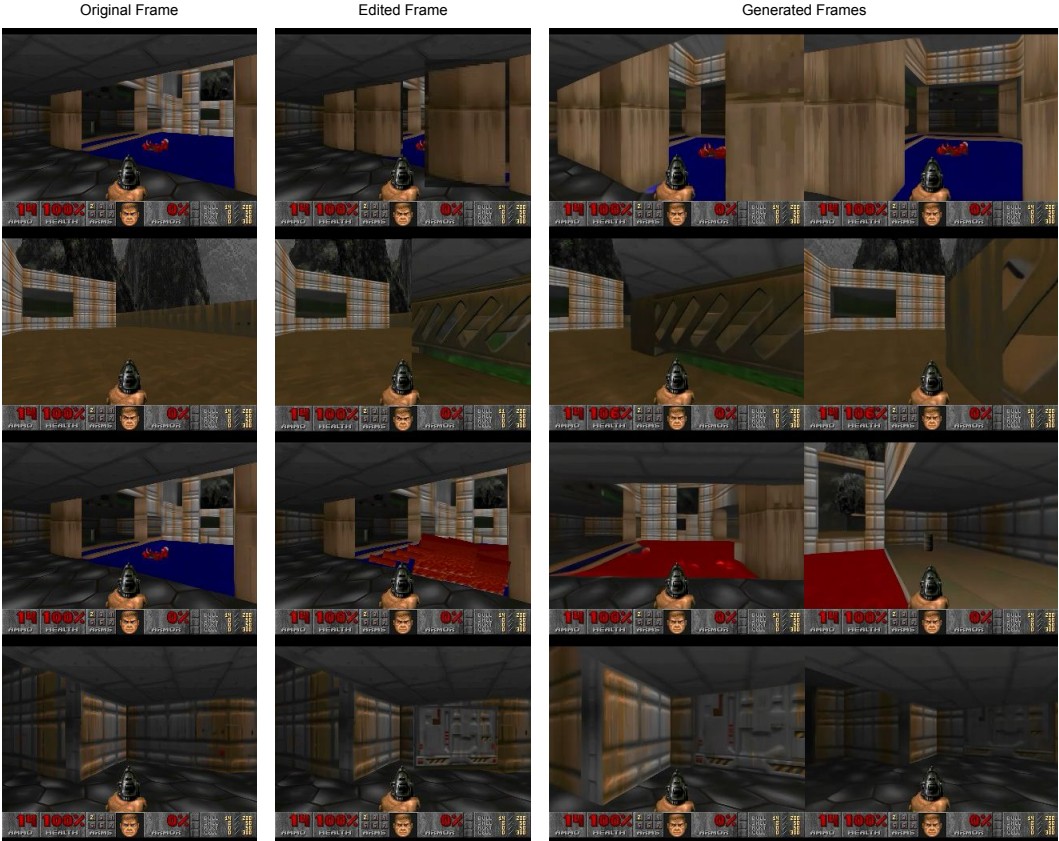

Figure 15: **Changing Structures.** Frames generated by GameNGen when starting generation from a manually edited state that combine level layout features from different levels not present in the training data.

## A.5    REWARD FUNCTION

The RL-agent's reward function, the only part of our method which is specific to the game DOOM, is a sum of the following conditions:

1. Player hit: -100 points.

2. Player death: -5,000 points.

3. Enemy hit: 300 points.

4. Enemy kill: 1,000 points.

5. Item/weapon pick up: 100 points.

6. Secret found: 500 points.

7. New area: 20 * (1 + 0.5 * $L_1$ distance) points.

8. Health delta: 10 * delta points.

9. Armor delta: 10 * delta points.

10. Ammo delta: 10 * max(0, delta) + min(0, delta) points.

Further, to encourage the agent to simulate smooth human play, we apply each agent action for 4 frames and additionally artificially increase the probability of repeating the previous action.

## A.6    REDUCING INFERENCE STEPS

We evaluated the performance of a GameNGen model with varying amounts of sampling steps when generating 2048 frames using teacher-forced trajectories on 35FPS data (the maximal sampling rate allowed by ViZDoom, lower than the maximal rate our model achieves with distillation, see below). Surprisingly, we observe that quality does not deteriorate when decreasing the number of steps to 4, but does deteriorate when using just a single sampling step (see Table 1).

As a potential remedy, we experimented with distilling our model, following Wang et al. (2023) and Yin et al. (2024). During distillation training we use 3 U-Nets, all initialized with a GameN-Gen model: generator, teacher, and fake-score model. The teacher remains frozen throughout the training. The fake-score model is continuously trained to predict the outputs of the generator with the standard diffusion loss. To train the generator, we use the teacher and the fake-score model to predict the noise added to an input image - $\epsilon_{\text{real}}$ and $\epsilon_{\text{fake}}$. We optimize the weights of the generator to minimize the generator gradient value at each pixel weighted by $\epsilon_{\text{real}} - \epsilon_{\text{fake}}$. When distilling we use a CFG of 1.5 to generate $\epsilon_{\text{real}}$. We train for 1000 steps with a batch size of 128. Note that unlike Yin et al. (2024) we train with varying amounts of noise and do not use a regularization loss (we hope to explore other distillation variants in future work). With distillation we are able to significantly improve the quality of a 1-step model (see "D" in Table 1), enabling running the game at 50FPS, albeit with a small impact to quality.

## A.7 AGENT VS RANDOM POLICY

Figure 16 shows the PSNR values compared to ground truth for a model train on the RL-agent's data and a model trained on the data from a random policy, after 3 second of auto-regressive generation, for a short session of human play. We observe that the agent is sometimes comparable to and sometime much better than the random policy.

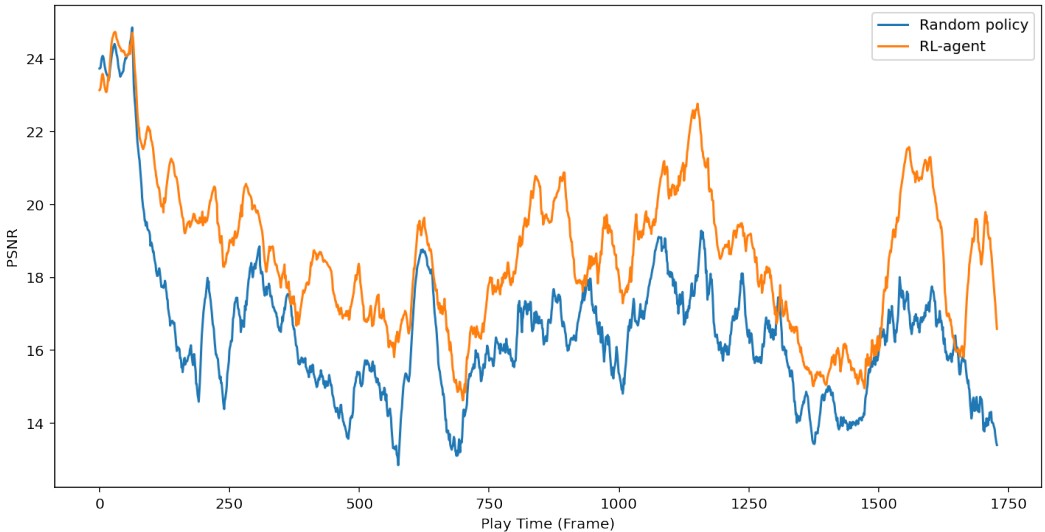

Figure 16: **RL-Agent vs. random policy over a short human play session.** Comparison of PSNR values between generated frame and ground truth for the agent (orange) and random policy (blue) after 3 second of auto-regressive generation. The values are smoothed with an EMA factor of 0.05.

## A.8 HUMAN EVAL TOOL

Figure 17 depicts a screenshot of the tool used for the human evaluations (Section 5.1).

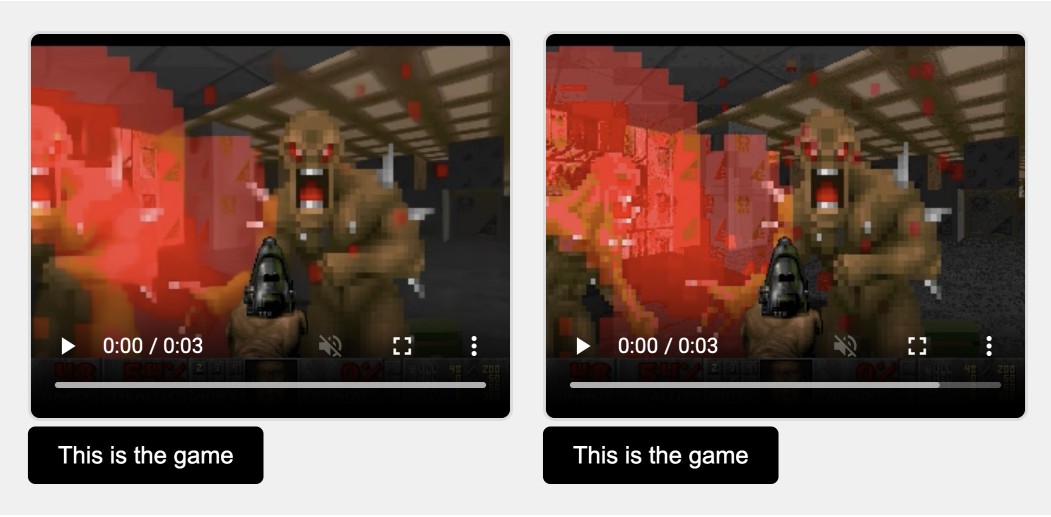

Figure 17: A screenshot of the tool used for human evaluations (see Section 5.1).

### A.9  ADDITIONAL RELATED WORK

**Interactive 3D Simulation.** Interactive 2D and 3D simulations are well-developed in computer graphics (Akenine-Mller et al., 2018). Game engines like Unreal and Unity a stream of images based on user input, tracking world state (e.g., player position, objects, lighting) and game logic (e.g., score). While film productions uses computationally intensive ray tracing (Shirley & Morley, 2008), game engines prioritize speed (30-60 FPS) with optimized polygon rasterization via GPUs. Physical effects, such as shadows and lighting, are approximated for efficiency rather than simulated with full accuracy.

**Neural 3D Simulation.** Neural methods for reconstructing 3D representations have made significant advances over the last years. NeRFs (Mildenhall et al., 2020) parameterize radiance fields using a deep neural network that is specifically optimized for a given scene from a set of images taken from various camera poses. Once trained, novel point of views of the scene can be sampled using volume rendering methods. Gaussian Splatting (Kerbl et al., 2023) approaches build on NeRFs but represent scenes using 3D Gaussians and adapted rasterization methods, unlocking faster training and rendering times. While demonstrating impressive reconstruction results and real-time interactivity, these methods are often limited to static scenes.

**Video Diffusion Models.** Diffusion models achieved state-of-the-art results in text-to-image generation (Saharia et al., 2022; Rombach et al., 2022; Ramesh et al., 2022; Podell et al., 2023), a line of work that has also been applied for text-to-video generation tasks (Ho et al., 2022; Blattmann et al., 2023b;a; Gupta et al., 2023; Girdhar et al., 2023; Bar-Tal et al., 2024). Despite impressive advancement in realism, text adherence and temporal consistency, video diffusion models remain too slow for real-time applications. Our work extends this line of work and adapts it for real-time generation conditioned autoregressively on a history of past observations and actions.

### A.10  SIMULATING A PLATFORM GAME

We experimented with the simple platform game "Chrome Dino" to demonstrate GameNGen's ability to simulate a different game type (see Fig. 18). For data gathering, we train an RL-Agent utilizing the Deep Q-Network (DQN Mnih et al. (2015a)) algorithm with experience replay, where the reward function is derived from the in-game score. The agent selects actions based on an epsilon greedy policy, with a decaying factor of 0.9995. During training, we recorded 2K episodes and used them to train the diffusion model using the same settings detailed in the main paper, with the following modifications: (1) we used a 32-frame context, (2) a resolution of 256x512, and (3) performed only 3,000 training steps. Similar to the results with DOOM, the simulation generated by GameNGen is fully playable over long trajectories in real-time, with visual quality comparable to the source game (see example video in the supplementary).

In addition, GameNGen supports simulating game-session management actions, such as game termination and automatic game replay. To achieve this, we concatenated two randomly selected episodes and sampled 32 frames from the combined sequence. This approach allows transitions between sessions to be represented when the sampled context spans both episodes.

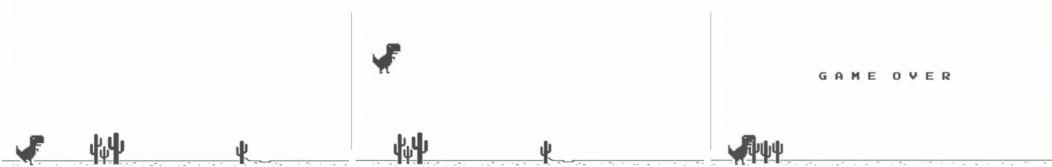

Figure 18: **Simulation of "Chrome Dino".** GameNGen automatically restarts the game session upon termination.

