# OpenReview forum: "Diffusion Models Are Real-Time Game Engines"
_ICLR.cc/2025/Conference — ICLR 2025 Poster_

### Official Review · Reviewer_Gte2 · 2024-10-30

**Soundness:** 3
**Presentation:** 3
**Contribution:** 3
**Rating:** 8
**Confidence:** 3

**Summary:**

The authors present a generative neural model which enables real-time controllable simulation of a complex game environment. This model is trained to reproduce Doom, using data gathered by training a separate RL agent in the VizDoom environment. They demonstrate that their model can produce high quality video output, and provide a variety of metrics to support this.

**Strengths:**

This is a clear, clean, well-presented paper, with impressive results. They have evaluated well, with interesting and relevant ablations and investigations. The chosen metrics seem well motivated and useful (LPIPS, PSNR, FVD). The desire for reproducibility is commendable and welcome, and I enjoyed the reverence they showed for Doom, having also “spent countless youth hours with the game”.

**Weaknesses:**

I think the paper’s biggest weakness lies in what the authors omit – I sense an understandable focus on what the model does _well_, and a glossing-over of what it might do _badly_. Some examples:

1. The human evaluation is run on clips of only 1.6 or 3.2 seconds. Why such short samples? Unless I missed it, no rationale is given for this, which leaves me to wonder whether the scores tail off significantly as the clips get longer.
2.  (Minor) The claim in the abstract that the image reconstruction is comparable to “lossy JPEG compression” becomes less impressive when it’s revealed, in section 5.1, to correspond to a JPEG quality setting of only 20-30.
3. Although it’s acknowledged (in 5.2.3) that certain map areas are “hard” or “easy”, there are no examples given of failure cases, or what it looks like when the model encounters a "hard" area.
4. Section 5.2.1 acknowledges that context length is problematic (and this is discussed in section 7), but the negative consequences of this aren’t spelled out – again, I’d like to have seen failure cases due to the lack of context. It felt as though the supplementary material was slightly cherry-picked to avoid this.


In the introduction, the authors posit the question _“Can a neural model running in real-time simulate a complex game at high quality?”_ and answer it with an unqualified “yes”. I recognise the impressive achievements in the paper, but this answer feels slightly dishonest – is the game genuinely playable (and enjoyable) in GameNGen, or are there still significant gaps to bridge? Can a level be played end-to-end? The PSNR scores are good, but they are not of themselves compelling evidence that the authors’ question can be answered in the affirmative. I think the paper would have been stronger if the answer to the posed question had been “Yes, _but_…”, or even “No, _but_…”

The second weakness in the paper is to do with _motivation_. Attempts to motivate the work by referring to a “new paradigm for game engines” feel a bit hand-wavy. Line 530 says that “the development process for video games under this new paradigm might be less costly and more accessible” – that’s a big “might”, when training GameNGen required 128 TPU-v5es, on top of training an RL agent to play the game in the first place. I find it hard to swallow any argument that a _new_ game can exist as weights of a neural model, rather than as lines of code: under the GameNGen paradigm, the game has to exist before the model can be trained. While the authors acknowledge that “many important questions remain”, I think it is possible to have a more grounded discussion around what these models can and can’t facilitate, and I would have liked to see this in the paper.

**Questions:**

Please consider these of lower priority – they are mainly for my own interest / to check my own understanding, and won’t necessarily have a bearing on my score.

1. I was very glad to see the ablations with the random agent, but it raised some questions:
  a) Does it follow similar practices to the RL agent (eg biased towards repeating last action; using each action for four time steps, as per A.5)?
  b) If not, could there be a significant distributional shift between the random actions and the human actions used for evaluation? Could this be compounding the low accuracy of the random model, alongside the problems caused by lack of exploration?
  c) Given the difference in model performance between the random data and the trained data, did the authors consider _not_ including the initial random exploration from the trained agent in the dataset, but instead training (or evaluating) the agent for longer to get a higher quality dataset?

2. What was the benefit of adding noise at _different levels_ during training time? And how was the noise level chosen at inference time?

3. How long did the training take? 128 TPU-v5e devices for how long?

4. A question about the numbers – the paper states that the RL agent ran for a total of 50M environment steps, but lines 295-6 state that the generative model was trained on a “random subset of 70M examples”.

5. The section on human evaluation was not entirely clear. To check my understanding: a set of 130 pairs of clips was generated, starting from random locations – pairing the ground-truth with the model output, but only running the model for 64 (or 32) steps. On these pairs, the test subjects could identify the model’s output 58% or 60% of the time. Then a _second_ set of 150 clips was generated by rolling out the model for 6000 or 12000 steps (five or ten minutes), and then capturing the next 60 frames? In which case, what were users asked to compare these with? Around line 317 it’s mentioned that the predicted and ground-truth trajectories diverge after a few steps, so presumably after ten minutes they are significantly different. So is the human then presented with two very diverse clips, and asked to work out which is the prediction? This would surely make the comparison much harder, since they would not be comparing like for like. Would this explain why they only scored 50% in this test? Apologies if I've misunderstood the setup.

6. Is there a sense of what the pre-trained Stable Diffusion 1.4 brings to the table? Did the authors experiment with end-to-end training, or using different pre-trained text-to-image models?

7. Fig 13 – the plot indicates that the 70M dataset PSNR metric would keep going up if trained for longer – did the authors try going beyond 700K steps?

---

> ### Author Response · Authors · 2024-11-21
>
> Thank you for your thoughtful comments. We’re thrilled that you found the paper well-presented, clear, and with impressive results. We put significant effort into making our evaluations comprehensive and included detailed implementation information to support the reproducibility of our work.
>
> We also want to thank you for your constructive review on limitations. It is important for us to not gloss over the weaknesses of our method, and we appreciate you raising this concern. While we have included our method’s limitations in the main text, and they are visible in the provided gameplay, we have added videos with explicit qualitative samples of failure cases to the supplementary including context issues and behavior in unexplored areas as you requested. In addition, we added a note in the eval section directing the readers to the longer videos provided in the supplementary for an example of the quality on longer time horizons (which is distinguishable from the original game but still playable and enjoyable).
>
> Regarding the phrasing of the introduction – we feel that the wording we used (“While not an exact simulation…”) is accurate; to be sure this is read as “yes, but” we’ve added a reference to the limitation section. Importantly, the simulation is playable and enjoyable, and the simple cpu-based agent we introduced is able to fully explore significant portions of the game (for example, the first level is playable until the last room).
>
> Regarding motivation, we do believe that neural game engines will enable new capabilities that were previously harder to achieve, and acknowledge that this is only partially explored in this work. For example, Appendix A4 provides a motivating example to how game behaviors can be developed from example images. Furthermore, it shows that a neural model is able to combine behaviors from different levels – making the argument that combining different behaviors from different games in order to create new ones may be possible as well. We are happy to continue working with you on the phrasing here as needed – our goal is to introduce possible research directions while acknowledging that are not fully explored in this work.
>
> Below we answer additional questions you’ve raised:
>
> **Can a level be played end-to-end?** The provided gameplay videos show the first level being played until the last room, taking ~1.5 minutes.
>
> **Short samples in the user-study**: It is true that longer examples are distinguishable from actual gameplay, but they are still playable and enjoyable. We added a note about this in the relevant section.
>
> **Random agent ablation**: The random agent repeats its actions for four time steps but it does not repeat last actions. There is indeed a significant difference between human actions and the random ones, however this is also the case for the agent-based data – when we tried increasing the bias for the last action too much, we saw that it prevents the agent from training well as it relies on stochastically exploring new behaviors. Training the agent for longer, or not including the initial random exploration is interesting to try – we hadn’t made these experiments.
>
> **Different levels of noise**: Adding different noise levels during training increases the robustness of the generative model. Higher noise levels encourage the network’s predictions to stay close to the training distribution, even under highly corrupted conditions, while low noise levels ensure that the model will cosider smaller details in the input frame. We ran training experiments where we only added low noise levels but found it performed worse than training on randomly corrupted conditions (both high and low corruption levels). During testing, we set the noise level condition to zero, conditioning on non-corrupted frames generated from previous game steps. We did not fully ablate this particular choice, and adding some artificial noise to the history may prove useful.
>
> **How long did training take**: 700k steps took 4 days.
>
> **50M env steps vs. 70M examples**: The number of environment steps the agent was trained on does not fully determine the amount of output as we also captured gameplay from large eval runs throughout the agent training.
>
> **Human eval**: The second set of clips is generated by rolling both the game and the simulation on 10 minute worth of actions. We compare clips between 5-10 minutes. It is true that after 5 minutes the game and the simulation diverge, so the clips look different.
>
> **Stable diffusion 1.4**: We haven’t ablated other architectures, nor training from scratch. We found Stable Diffusion 1.4 to be expressive and efficient for our task. We hope further development and better architecture design will be suggested in future works.
>
> **More than 700k steps**: We continued running for more steps (up to 1M) after the paper was written but did not run full evals. Indeed the PSNR continued to improve, and test loss went down.

---

> > ### Comment · Reviewer_Gte2 · 2024-11-25
> >
> > Thank you for your comprehensive reply, and for taking the time to answer my questions. The extra supplementary videos are well chosen, and clearly illustrate the limitations. I have no further questions, and I look forward to seeing the follow-up papers.

---

### Official Review · Reviewer_Ybe8 · 2024-11-01

**Soundness:** 3
**Presentation:** 3
**Contribution:** 2
**Rating:** 8
**Confidence:** 4

**Summary:**

This paper trains an action-conditional world model with diffusion for the environment of DOOM. The authors demonstrate that the model can be run in real-time at 20fps, and conduct ablations on the context length and necessity of noise augmentation during training to maintain stable generations.

**Strengths:**

Its fantastic that your model can run in real-time at 20+ fps. To me Section 3.3.2. is one of the most interesting parts of the paper. I would like to see this discussion expanded, and the important experiments/choices made moved from the appendix into the main paper. Especially since this is a substantial part of your contribution over other works.

This paper also provides more evidence to the community that learning world models of complex environments is entirely feasible.

I very much like the direction of the paper, and feel that it is presented well.
I do think this is good work that is of great interest to the community, but I cannot overlook the potential lack of novelty in relation to prior work given the current framing of the paper. Hence, the majority of my review is focussed on weaknesses.

**Weaknesses:**

# Major

## Prior work
The presentation of related/prior work is lacking in this paper.
There is substantial prior work in this domain which is either not mentioned or not correctly characterised.
In particular, "Diffusion for world modeling: Visual details matter in atari" cannot be considered concurrent work since it was first publicly available a year ago (in submission to last year's ICLR). Given the enormous similarities between your work and theirs, a much larger discussion is warranted - what is different/novel about your work compared to theirs, and more importantly highlight all of the similarities (being able to play in real-time, using diffusion for world modelling, architectural choices, etc). In addition, their paper utilises the model to train an RL agent, which makes progress towards addressing an important limitation you highlight about your work - "For example, our agent, even at the end of training, still does not explore all of the game’s locations and interactions, leading to erroneous behavior in those cases."

There is also no mention of GAIA-1, which simulates a complex real-world environment with a neural model.
Also no mention of "VideoGPT: Video Generation using VQ-VAE and Transformers" which also learns an action-conditional video prediction model for Doom.
"Temporally Consistent Transformers for Video Generation" also looks at long-term action conditional video generation quality.

No mention of "Diffusion Forcing: Next-token Prediction Meets Full-Sequence Diffusion" that also uses the same noise augmentation during training to improve long term autoregressive generation. (Yes technically ICLR considers this contemporary work since it was first posted to arxiv on 1 July 2024)

# Minor

"...GameNGen extracts gameplay..." - what does this mean, are you making a claim about the representations learned?

Figure 2 is unfair, the other papers were trained on very different data (notably without the HUD). At least try and use a comparable image when comparing your work to theirs. Given the higher resolution and increased visual fidelity of your model, there is no need to exaggerate the differences like this to highlight the improvements you've made.

Is there a need for a new definition of 'Interactive Environment' instead of using an existing formulation? Why doesn't a POMDP work, especially given your use of RL in the paper to generate the experiences.

# Post-rebuttal

Based on the replies I am happy to increase my score and recommend acceptance.

**Questions:**

Why do you start with a pre-trained text-to-image diffusion model, what are the motivations for doing so? Is there not enough data to train from scratch?

Data wise, what kind of coverage do you have of the game? How does your model perform in areas where there is comparatively little data? "When playing with the model manually, we observe that some areas are very easy for both, some areas are very hard for both, and in some the agent performs much better" - suggests you have conducted some initial investigations into this, please elaborate more on this.

Can you clarify exactly how much data you train on, is it 70M transitions? How many epochs of training does this correspond to? ~90M sequences (batch size of 128 with ~700k training steps) are trained on of length 64, so each timestep is seen roughly 64 times?

For your context length experiments, do you have any qualitative results or observations on utilising longer context lengths? If an object is present say 50 frames in the past, would the 64 context length correctly remember this whereas the 32 frame one wouldn't?

Given the open nature of the environment, will you be releasing code/data that could be used as a potential benchmark for future work in this area?

More discussion and examples on the long-term generations would be welcome (30 seconds/1 minute+). You mention that they are still hard to distinguish from real gameplay, but do they have a decent temporal consistency? Do the generations respect the geometry of the DOOM levels, do they always end up in specific areas, does the model count properly, etc.

---

> ### Author Response · Authors · 2024-11-20
>
> Thank you for your thoughtful review! We appreciate your insightful questions, positive feedback on the work’s relevance to the community, and help identifying key related works we missed. Below, we address issues you’ve raised.
>
> **More details on speed**: We agree and have moved the results into the main paper.
>
> **Related work**: We would love to work with you on improving this section and appreciate your feedback.
>
> "Diffusion for world modeling: Visual details matter in Atari" (Diamond): Thank you for highlighting the OpenReview version (named ["Diffusion world models"](https://bit.ly/3ZfaAIj)). The authors recently added a high-res Counter-Strike simulation comparable to our work, though without quantitative evaluations; earlier versions were low-res and hint at degeneration after a few frames [1]. We agree this work is highly relevant and have updated our paper to further highlight it.
>
> GameNGen achieves long trajectory, high-resolution simulation thanks to three key features not used by Diamond:
> - Agent-driven data collection: Counter-Strike in Diamond does not use an agent and therefore has limited training data. The Atari version aims at improving the agent and not the simulation. Therefore, the agent’s reward is not an exploration reward like the one we use. Moreover, even if the reward was geared towards exploration, training only in simulation prevents the agent from learning to explore elements that are not simulated yet (e.g. opening doors).
> - Noising the condition variable: Input noising prevents compounding errors when using the model autoregressively. Ablating noise from our model results in errors like in this [link](https://bit.ly/3B31fdf)).
> - Latent diffusion and optimizing the decoder - allows our model to render small details like text and faces at 20fps.
>
> We also added GAIA-1, VideoGPT, and Temporally Consistent Transformers for Video Generation and Diffusion Forcing to the paper. These are all important and relevant contributions and we thank you for highlighting them.
>
> **Concurrency**: We aim to include all relevant papers (concurrent or not) in our related work and appreciate your comments. For novelty, the ICLR FAQ considers publication at a *peer-reviewed* venue. The Diamond paper will appear at NEURIPS 2024 (December). A recent ArXiv update (October 30, 2024) added a high-res Counter-Strike simulation and cited GameNGen as concurrent work. We are fans of this effort (partially inspired by GameNGen [2]) and see it as a fruitful interaction between teams.
>
> **Extracting gameplay**: This phrasing highlights our method’s ability to use an RL agent to crawl an existing game and generate diverse, scalable training data.
>
> **Figure 2**: We included the HUD to highlight the challenge it adds to visual simulation (a naive approach with HUD fails; see Appendix A2). The caption now notes that other methods use different training data.
>
> **definition of 'Interactive Environment’**: We wanted to focus on the simulation aspect and our definition is similar to POMDP without rewards, as we don’t need them.
>
> **Pretrained vs. Starting from scratch**: We fine-tuned Stable Diffusion to leverage its visual knowledge and speed up training. While we didn't ablate this, starting from scratch would likely work but take longer to achieve similar quality.
>
> **Game coverage**: Agent exploration varies by starting point—for example, it fully explores the first level except the last room but struggles in areas requiring finding keys or killing many monsters. Sparse data can cause the model to hallucinate smooth transitions to well-explored areas (see unexplored_area.mp4 in the supplementary). Quantitative comparison is in Appendix A7 (e.g., frames 1500–1750).
>
> **Data**: The model is trained on ~70M examples (64 input frames and 1 target frame), randomly sampled from agent training sequences. Over 700k steps, each frame is reused ~45 times as input and ~1.3 times as a target.
>
> **Context length**: See limited_context.mp4 in the supplementary for a relevant example. As noted in the paper, context effects are typically short-range (e.g., objects, game physics, animation), with marginal improvement over longer ranges. This may be due to the rarity of cases like the one you describe in our training data and is an area for future work.
>
> **Releasing code**: We will look into releasing the code soon. Meanwhile, we used open-source tools and provided detailed implementation steps to allow for full reproduction by the community ([example](http://bit.ly/4hUL02D)).
>
> **Long generations**: See the supplementary (figure_1 folder) and videos [here](https://bit.ly/3ZhFO1i). The videos demonstrate temporal consistency, accurate level geometry, and correct weapon and health tracking in gameplay. More can be generated if needed.
>
> [1] from the paper: “Whilst in immediate frames these have the intended effect, for longer roll-outs the observations can degenerate.”
>
> [2] in a tweet by the author omitted for anonymity.

---

> > ### Comment · Reviewer_Ybe8 · 2024-12-02
> > **Thanks for your reply**
> >
> > (Apologies for the late reply)
> > Your replies to myself and the other reviewers (as well as the updates made to the paper) provide some useful and interesting information, please do consider incorporating it into the paper (such as each frame being reused ~45 times as input).
> >
> > I am happy to raise my score and recommend acceptance.

---

### Official Review · Reviewer_cdyX · 2024-11-02

**Soundness:** 3
**Presentation:** 3
**Contribution:** 2
**Rating:** 5
**Confidence:** 3

**Summary:**

This paper propose GameNGen, which is a diffusion model to predict the next frame given past observations and actions of a video game and serves as the game engine.

GameNGen runs in 20fps and achieves good quality of next frame prediction on the game VisDoom. Extensive experiments show that the frame generation in the autoregressive way can maintain important elements of the game UI.

**Strengths:**

1. The first work focus on interactive playable real-time simulation, interesting idea.
2. Extensive experiments and broad ablation study shows the accurate prediction (at least visually) of the diffusion model and also efficiency of some design choice.

**Weaknesses:**

I agree a neural simulator is an interesting idea, but it would be good to show more things: (just as what you mentioned as your future work)

1. Same method but generalized to more than one game, otherwise it might be suspicious that VisDoom has some aspect to be easy to learn (like the unchanged UI).

2. Shows how a neural game simulator can be useful for downstream tasks like using it to train an agent with faster speed or empower the agent with a great forward model to help decision making.


Hope your further results can enrich the paper to be above the acceptance threshold.

**Questions:**

none

---

> ### Author Response · Authors · 2024-11-20
>
> Thank you for reviewing our paper and providing valuable feedback to help us improve it.
>
> **Regarding using Doom**: The focus on DOOM was driven by the desire for reproducibility (as it is an open-source game engine), while still providing complex game logic and graphics. It’s a good point that some aspects of Doom may have helped, we tried to highlight these in the limitations section (we’re revising the text to make this clearer - thank you). However, the game is sufficiently complex to serve as a proof of our concept, including many ways to interact with the environment (open doors, pick up armor, explode barrels, attack enemies, walk on lava), as well as diverse geometry and visual details.
>
> **Regarding downstream tasks**: The agent-related tasks that you mentioned would be an interesting follow up (we’re inspired by the many important works here like [1, 2, 3]). Our work focuses on the relatively less-explored area of using neural models as game engines. In this context, a critical downstream application is the ability to generate new levels and behaviors seamlessly. Appendix A4 presents preliminary results demonstrating this capability, where new behaviors are generated using input images created with a graphical editor such as Photoshop. These results are particularly exciting as they highlight a potential advantage of neural game engines over traditional ones: the ability to guide game behavior simply by generating a static scene with a graphical editor. Please also see the supplementary material (folder appendix_A4_edit_game) for relevant videos illustrating these results. Would this address your request for additional downstream tasks?
>
> [1] Ha, David, and Jürgen Schmidhuber. "Recurrent world models facilitate policy evolution." Advances in neural information processing systems 31 (2018).
>
> [2] Kaiser, Łukasz, et al. "Model Based Reinforcement Learning for Atari." 8th International Conference on Learning Representations, ICLR 2020.
>
> [3] Danijar Hafner, Timothy Lillicrap, Jimmy Ba, and Mohammad Norouzi. “Dream to control: Learning behaviors by latent imagination”. 8th International Conference on Learning Representations, ICLR 2020.

---

> > ### Comment · Reviewer_cdyX · 2024-11-26
> > **not change the score**
> >
> > Thank you for your reply. After carefully reading them, I decide to not to change my score.

---

> > > ### Author Response · Authors · 2024-11-26
> > >
> > > Thank you again for your thoughtful suggestions in your original review. As noted in our previous response, we initially focused on Doom for its complexity and reproducibility. However, we also recognize the importance of demonstrating the method’s generalizability across different games, and we thank you for pointing this out. To address this, we applied GameNGen on a simple platform game, achieving a fully playable simulation, capable of supporting long trajectories in real time with visual quality comparable to the source game.
> > >
> > > We used exactly the same method detailed in the paper (except using history depth of 32 frames) with a different agent optimized for the game. We only used 2k episodes and trained for 3k steps. Since the game is simple, this is enough to reach a playable version.
> > >
> > > We have detailed this experiment appendix A9 and included a video showcasing a long, human-played trajectory in the supplementary materials  (chrome_dino_gameplay.mp4). We hope this addition addresses your concern, and would be grateful to know otherwise so that we can further improve our work.

---

### Official Review · Reviewer_JQcB · 2024-11-02

**Soundness:** 4
**Presentation:** 4
**Contribution:** 3
**Rating:** 8
**Confidence:** 4

**Summary:**

This paper demonstrates for the first time that DOOM, a classic first-person shooter game, can be simulated in real time by an action-conditioned video diffusion model. The paper first collects gameplay trajectories using RL where the reward is to mimic human gameplay footage. Then, the paper trains an action-conditioned video diffusion model that, given recent gameplay frames and actions, generates the next video frames. Model behavior and effects of different hyperparameters are thoroughly analyzed.

**Strengths:**

- The paper demonstrates for the first time that one can make a neural network real-time simulate a relatively complex video game. The motivation, rapid text or image-programmable video game generation, is clear and convincing. I appreciate the amount of engineering that went into making this, which seemed far-fetched a year or two ago, happen.
- The paper provides a plethora of metrics from PNSR, LPIPS, FVD, and human evaluations on model-generated image and video quality.
- The paper provides comprehensive ablations on hyperparameter choices like context length, noise augmentation of the conditioning variables, and gameplay data.

**Weaknesses:**

- There is no methodological novelty to the paper, but given the remarkable findings this is not a problem.
- The model and code are not available to the public, so we cannot assess how robust the model and generated gameplay is. Since this is a phenomenological paper, this is more important than it is for typical ML papers.
- It is unclear how much of this amazing performance is due to "training data overfitting", and how well the model would perform on a sufficiently different DOOM map. The authors mention that the model is able to memorize map structure for much longer than its context window, which spans seconds. My guess is that this because the model is overfitting the training DOOM map. While the authors visually investigate what happens when enemies from later levels are introduced earlier in the game (OOD setting), having quantitative metrics in such settings would make the paper stronger.
- Related work section should include prior work that adds noise to conditioning variables, for example Ruhe et al [1].

[1] Ruhe, D., Heek, J., Salimans, T., & Hoogeboom, E. (2024). Rolling Diffusion Models. arXiv preprint arXiv:2402.09470.

**Questions:**

- How well do you think the model will generalize to unseen DOOM gameplay (ex. custom maps)?
- The model was able to generate short to medium gameplay footage that was hard to distinguish whether they were model-generated for normal humans when noise augmentation on the conditioning variables was applied. If conditioning variable noise augmentation is not applied, do you think this fact will change?

---

> ### Author Response · Authors · 2024-11-20
>
> Thank you for your thoughtful comments and positive feedback. We appreciate your recognition of the significance of demonstrating that a neural network can simulate a relatively complex video game in real time. We also felt that this seemed far-fetched not long ago—and appreciate your acknowledgment of the engineering effort involved. We're particularly glad you found value in the diverse metrics and ablations, as it was important for us to fully evaluate and validate our approach.
>
> We changed the paper to include the related work that we have missed in the main text. Please see below answers to specific questions raised by you.
>
> **Model Generalization to unseen gameplay**: The model indeed memorizes the structure of the areas it was trained on, but it is able to simulate new trajectories, viewpoints and situations in these areas. When the model is initialized to a completely different unseen area, it will gradually transition it to a familiar area (see unexplored_area.mp4 in the supplementary for an example). In other, less extreme, cases (as seen in appendix A4) the model is able to combine new elements that did not appear in the training data into the simulation.
>
> **Code and model release**: Unfortunately we are unable to release the model weights at this time but we are hoping to be able to do so or at least release the code soon. Regardless, we’ve made the training recipe as detailed as possible and used open source base models hoping that the community creates an open source reproduction ([example](http://bit.ly/4hUL02D)). We’ve also added gameplay examples, including failure cases (added in the latest version of the supplementary), to further help readers evaluate the model performance.
>
> **Importance of noise augmentation**: Without noise augmentation the simulation deteriorates quickly, especially when the player does not keep moving (see Fig. 4 for an example).

---

> ### Comment · Reviewer_JQcB · 2024-11-21
>
> Thank you for your reply. I will maintain my score trusting that the authors will release the code soon. While GameNGen's amazing performance is in part due to training data overfitting, I still believe that GameNGen is an impressive step forward from the prior neural game engine-related work.

---

### Official Review · Reviewer_FBWE · 2024-11-03

**Soundness:** 2
**Presentation:** 2
**Contribution:** 2
**Rating:** 5
**Confidence:** 3

**Summary:**

This paper introduces "GameNGen," a real-time game simulation engine using a neural diffusion model to mimic the gameplay of DOOM. GameNGen is claimed to be a first in simulating interactive environments with high fidelity and extended play sessions, using stable diffusion-based architectures. The approach involves a reinforcement learning (RL) agent collecting game data, which is then fed into a diffusion model to train on successive game frames. The authors report that their model achieves near-indistinguishable results from actual gameplay in short clips and can maintain consistency in long sequences.

**Strengths:**

Real-Time Performance: The paper demonstrates a model that runs at 20 frames per second, achieving performance close to real-time gaming on a TPU, which shows its practical deployment potential in high-demand applications.

**Weaknesses:**

Lack of Novelty: The application relies on pre-existing models, primarily a stable diffusion variant, with incremental architectural adjustments. While the use of diffusion models in gaming is somewhat novel, the approach is more of an adaptation than a breakthrough innovation in game simulation.

Clarity: I found Section 2 difficult to understand. Could you please elaborate on the model inputs and clarify the regression objective? The mathematical symbols are a bit confusing—for example, could you explain what \(o_{q_i}\) and \(o_{p_i}\) represent?

Limited Scope of Application: This work is demonstrated solely on DOOM, an older game with relatively simple graphics. The approach may face challenges when applied to modern, complex games with higher resolutions, demanding more model robustness and memory capacity.

**Questions:**

Please see the weakness.

---

> ### Author Response · Authors · 2024-11-20
>
> Thank you for reviewing our paper and providing thoughtful feedback. Please find our replies to your feedback below.
>
> **To the point of novelty**: we see the novelty and contribution of the work in introducing the task of real-time interactive game simulation, and in the successful adaptation of a widely available open-source text-to-image model (Stable Diffusion) to it. Apriori, the fact that it is possible to simulate a complex game in real-time with high-res details (like text) using a diffusion model, was not immediately obvious [1]. We provide quantitative evidence for simulation quality, detailed ablations of key components, and a reproducible training recipe, which we hope are valuable contributions to the ICLR community.
>
> At the level of technical novelty, we would like to highlight that straightforward application of text-to-image models does not work, and three technical innovations were needed:
> 1. We found noise-augmentation to be critical to make auto-regressive sampling work over a long time horizon. To the best of our knowledge, this method was not previously applied in an auto-regressive context.
> 2. Collection of training data using an RL agent was required for scaling data beyond human-play recordings.
> 3. Decoder fine-tuning for achieving comparable visual quality as the source game.
>
> **Clarifying Section 2**: Thank you for your questions and apologies if the notation in this part was not clear enough. The simulation model $q$ gets as inputs an history of observations $o_{<n}$ and actions $a_{
> \le n}$, and produces the next observation $o_n$. Note that we have two streams of observations, one coming from the model ($o_q^i$) and another coming from the ground truth environment ($o_p^i$). The input to the simulation can either be observations that were obtained auto-regressively from the model itself, or the ground truth observations. While training we use the latter (we refer to this as “teacher forcing” in the paper).
> We define the task of “Interactive World Simulation” as minimizing some given distance metric D between the two sequences mentioned. We’re trying to see if we can amend the writing slightly to make this clearer and please let us know if you have further questions or suggestions on how to improve the notation here.
>
> **Scope of Application**: The focus on DOOM was driven by the desire for reproducibility (as it is an open-source game engine), while still providing complex game logic and graphics. The game includes many ways to interact with the environment (open doors, pick up armor, explode barrels, attack enemies, walk on lava), as well as diverse geometry and visual detail. It’s a good point that the photorealism of DOOM is lower than modern games. In terms of game complexity we see it as very challenging for simulation, especially given that many prior work in this space focused on simpler environments such as Atari games.
>
> [1] For example see this quote from JqCB: “I appreciate the amount of engineering that went into making this, which seemed far-fetched a year or two ago”.

---

### Meta-Review · Area_Chair_nRpd · 2024-12-21

**Metareview:**

After careful consideration of the five expert reviews, the subsequent author-reviewer discussions, and the area chair’s own reading of the paper, I recommend accepting this submission. The paper presents an interesting work in using diffusion models for real-time game simulation, and demonstrates supportive results.

The paper's primary technical contribution centers on GameNGen, a system that can generate high-quality, interactive game simulations at 20 frames per second. While the approach builds on existing diffusion model architectures, the authors have made several technical innovations to achieve stable, long-term generation, including noise augmentation during training and decoder fine-tuning for visual detail preservation.

The experimental validation is thorough. The authors provide quantitative metrics (PSNR, LPIPS, FVD) and human evaluations that demonstrate the quality of their generated content. They also include detailed ablation studies that justify their design choices and show the impact of different components. The reviewers appreciated the authors' commitment to reproducibility, providing detailed implementation information and using open-source tools where possible.

Two reviewers (FBWE and cdyX) expressed concerns about the novelty and scope of the work, noting its reliance on a single game environment and existing model architectures. However, these reviews were relatively brief and did not engage deeply with the technical contributions. In contrast, three reviewers (JQcB, Ybe8, and Gte2) provided thorough technical analyses that recognized the significant engineering achievements and broader implications of the work.

During the discussion period, the authors engaged constructively with reviewer feedback, and provided additional results and clarifications. They added failure case examples to the supplementary material, expanded their evaluation to include a simpler platform game, and enhanced their discussion of limitations.

The paper does have limitations, particularly in its focus on a single main game environment. However, the extents of the evaluation, the real-time performance achievement, and the potential impact on future game development make these limitations acceptable. The authors have also been appropriately transparent about these constraints and their implications.

**Additional Comments On Reviewer Discussion:**

During the rebuttal period, there was extensive and constructive discussion between the authors and reviewers regarding several key aspects of the paper. The reviewers' feedback focused on three main areas: technical comprehensiveness, method generalizability, and result validation.

Reviewer Ybe8 raised concerns about missing citations and insufficient discussion of related work, particularly regarding prior work on diffusion models for world modeling. The authors addressed these points by expanding their literature review and clarifying the technical distinctions between their approach and previous work, especially regarding the “Diamond” paper. After the authors' response, this reviewer increased the score to 8 and recommended acceptance.

Reviewer cdyX questioned the method's generalizability beyond DOOM and requested demonstration of downstream applications. The authors responded by implementing their approach on a simpler platform game and highlighting their preliminary results on level editing capabilities. While this reviewer maintained their original score of 5, their concerns were relatively high-level compared to the more technical reviews.

Reviewer Gte2 provided detailed feedback about potential limitations and requested more explicit discussion of failure cases. The authors responded comprehensively by adding failure case examples to the supplementary material and enhancing their discussion of limitations. This reviewer maintained the strong positive assessment of the work.

The most substantive discussion occurred around technical innovation versus engineering achievement. While some reviewers initially questioned the novelty, the authors effectively demonstrated their technical contributions through detailed ablation studies and clear explanation of their innovations in noise augmentation and decoder fine-tuning.

In weighing these discussions, the detailed technical responses and additional experimental results provided by the authors effectively addressed the main concerns raised by the reviewers. The fact that two of the less favorable reviews were relatively brief and focused on high-level concerns, while the positive reviews provided in-depth technical analysis, further supported the decision to accept the paper.

---

### Decision · Program_Chairs · 2025-01-22

Accept (Poster)